# Circum-Mediterranean influence in the Y-chromosome lineages associated with prostate cancer in Mexican men: A Converso heritage founder effect?

Esmeralda Álvarez-Topete[1⊚], Luisa E. Torres-Sánchez[2⊚], Esther A. Hernández-Tobías[3], David Véliz[4], Jesús G. Hernández-Pérez[2,5], Ma. de Lourdes López-González[1], Marco Antonio Meraz-Ríos[6], Rocío Gómez[1]*

1 Departamento de Toxicología, CINVESTAV-IPN, Mexico City, Mexico, 2 Centro de Investigación en Salud Poblacional, Instituto Nacional de Salud Pública (INSP), Cuernavaca, Morelos, México, 3 Universidad Autónoma de Nuevo León, Facultad de Salud Pública y Nutrición, Monterrey, Nuevo León, Mexico, 4 Departamento de Ciencias Ecológicas, Instituto de Ecología y Biodiversidad (IEB), Facultad de Ciencias, Universidad de Chile, Santiago, Chile, 5 Escuela de Salud Pública de México, INSP, Cuernavaca, Morelos, México, 6 Departamento de Biomedicina Molecular, CINVESTAV-IPN, Mexico City, Mexico

⊚ These authors contributed equally to this work.
* mrgomez@cinvestav.mx

**Citation:** Álvarez-Topete E, Torres-Sánchez LE, Hernández-Tobías EA, Véliz D, Hernández-Pérez JG, de Lourdes López-González M., et al. (2024) Circum-Mediterranean influence in the Y-chromosome lineages associated with prostate cancer in Mexican men: A Converso heritage founder effect? PLoS ONE 19(8): e0308092. https://doi.org/10.1371/journal.pone.0308092

## Abstract

Prostate cancer is the second most common neoplasia amongst men worldwide. Hereditary susceptibility and ancestral heritage are well-established risk factors that explain the disparity trends across different ethnicities, populations, and regions even within the same country. The Y-chromosome has been considered a prototype biomarker for male health. African, European, Middle Eastern, and Hispanic ancestries exhibit the highest incidences of such neoplasia; Asians have the lowest rates. Nonetheless, the contribution of ancestry patterns has been scarcely explored among Latino males. The Mexican population has an extremely diverse genetic architecture where all the aforementioned ancestral backgrounds converge. Trans-ethnic research could illuminate the aetiology of prostate cancer, involving the migratory patterns, founder effects, and the ethnic contributions to its disparate incidence rates. The contribution of the ancestral heritage to prostate cancer risk were explored through a case-control study (152 cases and 372 controls) study in Mexican Mestizo males. Seventeen microsatellites were used to trace back the ancestral heritage using two Bayesian predictor methods. The lineage R1a seems to contribute to prostate cancer ($OR_{adjusted}$:8.04, $_{95\%}$CI:1.41–45.80) development, whereas E1b1a/E1b1b and GHIJ contributed to well-differentiated (Gleason $\leq$ 7), and late-onset prostate cancer. Meta-analyses reinforced our findings. The mentioned lineages exhibited a connection with the Middle Eastern and North African populations that enriched the patrilineal diversity to the southeast region of the Iberian Peninsula. This ancestral legacy arrived at the New World with the Spanish and Sephardim migrations. Our findings reinforced the contribution of family history and ethnic background to prostate cancer risk, although should be confirmed using a large sample size. Nonetheless, given its complex aetiology, in addition to the genetic component, the lifestyle and xenobiotic exposition could also influence the obtained results.

**Data Availability Statement:** All relevant data are within the paper and its Supporting Information files.

**Funding:** Here are the CONACYT grant numbers involved in our paper: 261268 and 178239 (to RG), and 140482 and 272810 (to LETS). "The funders had no role in study design, data collection and analysis, decision to publish, or preparation of the manuscript."

**Competing interests:** The authors have declared that no competing interests exist.

## Introduction

Prostate cancer (PrCa) has shown a rapid increase in its incidence of almost five percentage points worldwide in the period from 1990 to 2019 [1]. In Mexico, PrCa is a leading cause of incidence of cancer in men, exhibiting heterogeneous patterns. Overall, the highest incidences are found in North and West Mexican states, decreasing southward [2]. Amongst the broad panorama of risk factors, cardiometabolic diseases, long-term high testosterone levels, age, lifestyle (i.e., diet, smoking and physical activity), and environmental factors have been associated with the PrCa risk [3]. The genetic factors are cornerstones of PrCa development, contributing between 30% and 45% [4]. Family history of breast cancer and/or PrCa have revealed positive correlations with high heritability rates [5]. Variable trends are not only evident across regions but also globally, as ethnic incidence disparities have also been reported, showing differences of up to 50 times amongst ethnicities [6]. European ancestry has shown the highest age-standardised incidence rates (ASIRs) per 100,000 inhabitants in a range of 46.76 to 78.67, followed by North African and Middle Eastern countries (range: 12–74) [1, 7]. Sub-Saharan Africa has presented ASIRs ranging from 26.56 to 56.13 [1]. The Americas exhibit a broad panorama; North American high-income countries have the highest ASIRs (113.02), whereas Latin American countries presented a range between 47.52 and 62.29 [1]. The lowest ASIRs have been reported within the Asian ancestry (range of 9.26–31.64) [1].

The Y-chromosome is a haploid genome and, in turn, is inherited uniparentally. Nearly 95% of this chromosome does not recombine (non-recombining region of the Y-chromosome, NRY), which is used to reconstruct the populations' genetic history [8, 9]. Through standardised terminology, more than 150 lineages (haplogroups) have been described, allowing for identification of the anthropological movements that shape the genetic heritage of populations [10]. The Mexican genetic architecture is particularly complex and regionally heterogeneous. Matrilineally, almost 89% of the lineages are Native Americans, approximately 2% originated from Africa, and 8% from West Eurasia [11]. The patrilineal landscape is integrated by the Native American genetic wealth (Q-MEH2, Q-L54, and Q-M3) [8]. Such lineages present the highest frequencies within the Native Americans (~0.760), whereas, in Mestizos, they are ≤ 0.350, following a clinal distribution increasing southward [9]. The European legacy also shapes the genetic landscape through the sex-biased miscegenation with the Iberian conquerors [12, 13]. Such ancestry has mainly been represented by the haplogroups E, G, I, J, L, R, and T, each with its own intrinsic diversity [14–16].

### From the Iberian southwest region to Mexico: The Circum-Mediterranean patrilineal legacy

The Iberian Peninsula has a wealthy genetic reservoir resulting from continuous migratory movements that has shaped its gene pool. The southwest region of Spain (Andalusia and Extremadura) underwent remarkable economic, historical, and theological events that had an impact on the Mexican genetic reservoir [9]. This geographic region is bounded by Portugal to west, the Mediterranean Sea to the southeast, and is separated from North Africa by approximately 15 kilometres of through the Strait of Gibraltar. Its proximity to the Mediterranean Sea, intensified the gene flow through the trade routes with the Maghreb (present-day Algeria, Libya, Mauritania, Morocco, Tunisia, and Western Sahara) [9, 17]. The commercial channels followed by Greeks, Romans, and Phoenicians, along with the continuous gene flow from the Levant (modern Cyprus, Israel, Jordan, Lebanon, and Syria) also contributed to delineating the Iberian genetic landscape [9, 18–20]. Archaeological and genetic evidence have suggested a bidirectional interaction between the Iberian southern populations and the North Africans

since the Last Neolithic [20]. North Africa was also a land bridge for migrations from the Arabian Peninsula, further enriching the Iberic genetic landscape [19].

The Islamic domain and the Semitic-speaking populations' diaspora shaped its genetic variation [9, 16]. The prominent representatives of the Abrahamic religions (i.e., Christianity, Islam and Judaism) converged in this southwest Iberian region, as well as in other Spanish regions. The Roman Empire founded *Hispania* (206 Christian Era, CE), and with them arrived Semitic migrants, possibly as either enslaved people or merchants [9]. Other Middle Eastern populations could have also come with the Jewish people to the Iberia, whose settlements might have occurred around 70 to 390 CE [16, 21]. Since the first century *Anno Domini* (AD), Jewish populations were dispersed across the Mediterranean Sea, and later, they settled down in the northern part of the Peninsula [22]. People whose descendants were Jews inhabiting Spain or Portugal were named Sephardim from Sefarad, the Hebrew word for Spain [23]. Around 711 CE, the *Al-Andalus* kingdom was established by the Islamic conquerors who occupied the Iberian territory for 800 years [20]. Religious tolerance was a signature of the Islamic period; Christians, Jews, and Muslims co-existed (*Convivencia* period) during the fourteenth and fifteenth centuries [19, 21]. Finally, during the *Reconquista* of the Christian forces, Jews (also known as *Marranos*) and Muslims (also known as *Moriscos*) were forcibly converted to new Christianity (mainly Catholicism), adopting new surnames [16, 19, 21]. Certain Semitic groups continued practising their religious adherence, such as the Crypto-Jewish and Crypto-Muslims secretly [19, 21]. During the Inquisition, which began first in Spain (1478) and then in Portugal (1536), fairly close to 200,000 Sephardim were relocated from Spain to the Mediterranean area; at least 100,000 of them to Portugal [16, 19, 21].

The Y-chromosome has been considered a biomarker prototype in terms of male health. Multiethnic populations provide a *sui generis* opportunity to enhance our understanding of the impact of specific ancestries on particular health risks along with the possibility of detecting founder effects [12]. The Y-chromosome has been previously associated with cardiovascular, infertility, and neurological disorders, as well as in male-specific neoplasia such as PrCa [24]. Genetic susceptibility and ancestral heritage are well-established risk PrCa factors [25]. Hence, the paternal origin of 152 incident PrCa cases and 372 population controls from the Mexican Mestizo population were traced back in the current study to determine its contribution to PrCa risk.

## Materials and methods

### Eligibility criteria

This study employed a randomised subsample taken from a previous case-control study [26]. Participants were unrelated Mexican Mestizo men between 42 and 94 years old, with at least one year of residence in Mexico City (CDMX). None of the participants presented prior history of any other type of cancer. Briefly, from November 2011 to August 2014, 402 incident cases of PrCa were identified, and confirmed histologically. Based on Gleason's score, PrCa cases were classified according to the tumour histological differentiation: 1) poorly (Gleason $\geq 8$), 2) moderately (Gleason = 7), and 3) well-differentiated (Gleason $\leq 6$) [27]. About the age at diagnosis, the cases were classified as early-onset PrCa (EOPrCa; $\leq 60$ years old) and late-onset PrCa (LOPrCa; $> 60$ years old). Each case was age-matched with two controls ($\pm 5$ years) without any history of PrCa and symptomatology associated with such pathology (i.e., dysuria and haematuria, and prostate-specific antigen $< 4$ ng/mL). In cases and in controls, the participation rates were 85.9% and 87.5%, respectively.

Once the participants received an explanation about their involvement and signed the informed consent form, an interview (average duration of 45 min) was conducted by trained

personnel using a structured questionnaire. The questionnaire included sociodemographic features (i.e., birthplace, education, marital status, and occupation), PrCa's family history, and lifestyle PrCa's risk factors (i.e., diet, physical activity, and tobacco consumption). Data from birthplaces was distributed into four geographic regions (Central, East, North, and South). The Central region was subdivided into Central-East comprising the states of Hidalgo, Estado de México, Morelos, Puebla, Querétaro, and Tlaxcala; Central-West including the states of Aguascalientes, Colima, Guanajuato, Jalisco, and Michoacán; and Central Valley represented by Mexico City. The East region included the states of Tabasco and Veracruz. In turn the Northern region included the states of Chihuahua, Coahuila, Nayarit, San Luis Potosí, Sinaloa, and Zacatecas, whereas the Southern region comprised the states of Chiapas, Guerrero, Oaxaca, and Yucatán. The present study included 152 PrCa cases and 372 controls randomly selected among those DNA samples available.

This study was conducted in accordance with the principles outlined in the Declaration of Helsinki. The Ethics Committee of the Mexican National Institute of Public Health (acronym in Spanish, INSP; CI:980) and other participating institutions approved the data and sample collection protocol for the current project.

## Molecular genetic analyses

**DNA isolation.** Peripheral venous blood samples (~7mL) were collected using the Vacutainer system (Becton Dickinson, Franklin Lakes, NJ, USA), and transported every day at 4°C to the INSP. DNA was isolated from the buffy coat (compiled with Ficoll-hypaque; Sigma Aldrich, St. Louis, MO, USA) with TRIzol (Thermo Fischer Scientific Inc, Waltham, MA, USA). A microvolume spectrophotometer Nano-Drop 1000 (Thermo Fischer Scientific Inc, Waltham, MA, USA) was used to determine the DNA yield and purity. The integrity was assessed using electrophoresis using 0.8% agarose gels.

**Genotyping.** All samples were genetically characterised with seventeen Y-chromosome short tandem repeats (YSTRs) located in the NRY with AmpFℓSTR™ Yfiler PCR amplification kit (Thermo Fischer Scientific Inc, Waltham, MA, USA) using a Verity 96-Well Fast Thermal Cycler (Thermo Fischer Scientific Inc, Waltham, MA, USA). Amplicons were run on an ABI Prism 3130XL Genetic Analyser, and fragment sizes were mapped with the GeneMapper ID v.3.2 software (Thermo Fischer Scientific Inc, Waltham, MA, USA) following the manufacturer's instructions. Control DNA 007 (Thermo Fischer Scientific Inc, Waltham, MA, USA) was analysed simultaneously to ensure the correct allele assignment. The sample characterisation was blinded; the processor did not know the type of sample (case or control) for genotyping.

## Lineage assignment

Haplogroup assignments were made with the 17-YSTRs haplotypes using two different Bayesian-Allele-Frequency predictor software (http://www.hprg.com/hapest5/; https://www.nevgen.org/) [28]. Only those haplotypes assigned with at least a 70% probability were included in the remaining analyses. Y-chromosome Consortium-2008 and the International Society of Genetic Genealogy 2019–20 recommendations were considered to determine the haplogroup nomenclature [29].

## Statistical analyses

Haplotype diversity ($h$) and the mean pairwise differences (MPD) were estimated with Arlequin v3.5 using 1,000 permutations [30]. Genetic distances ($R_{ST}$) were determined using an exact test of population differentiation with 10,000 steps in the Markov chain and 1,000 demorisation steps with Arlequin v3.5 software [30]. To avoid type I errors, the resulting $p$-values were adjusted using the false discovery rate with the function p adjust implemented in test

in R software (R Core Team 2023); $p$-values $< 0.05$ were considered significant [31]. $R_{ST}$ values were visualised in a multidimensional scaling plot (MDS) implemented in SPSS v11 (IBM Statistical Package for the Social Sciences for Windows, Chicago, Inc).

Cases and controls were compared based on specific features; chi-square or t-student testing was applied depending on the data type. An unconditional logistic regression model adjusted for age and family history of PrCa was used to analyse the associations between the lineages and PrCa; haplogroup Q was used as a reference group. All the analyses were carried out with Stata v14.0 software (StataCorp. 2015. Stata Statistical Software: Release 14.0. College Station, TX: StataCorp LLC).

## Comparison with other populations

Comparisons were made with populations that contributed to the Mexican Mestizo genetic architecture, including African, European, Middle Eastern, and Native Americans; NAM) were made in haplogroups exhibiting differences between cases and controls. Further, other Mexican mestizo and Iberian Peninsula (IBP) haplotypes were included.

A linear discriminant analysis was conducted to quantify the relationship between cases and controls with other populations. Such analysis maximises the differences between object groups (i.e., individuals clustered in populations) based on a set of variables (i.e., NRY microsatellites loci). The resulting discriminant equations also enable the assignment of new objects (i.e., cases and controls) to any population groups depending on the similarity degrees measured as probability. The analysis was conducted using the Modern Applied Statistics with S (MASS) package implemented in R software [32].

With the haplotypes previously assigned, network analyses were made. The phylogenetic relationships within the haplotypes of each haplogroup were built with Network v.10.0 (http://www.fluxus-engineering.com/) using the median-joining (MJ) option and visualised with Network Publisher software [33]. DYS385a/b locus was excluded from MJ networks because it represents a duplicate STR *locus*. DYS839 allele assignment was determined by the difference between DYS389II and DYS389I. All *loci* were weighted with the inverse of the variance.

Our findings were compared with analogous published data obtained from the PubMed database. Haplotypes for which haplogroup assignments were not reported were obtained with the two previously mentioned software programs. Furthermore, our findings were compared with the published data where some PrCa risk had been reported in carriers of specific NRY lineages. The search strategy included free-text terms such as "haplogroup", "lineage", "Y-chromosome", "microsatellites", and "polymorphisms" combined with the terms "prostate, cancer" and "prostatic, neoplasms". Meta-analyses were done using the web-based app "Meta-analysis made easy" developed with Shiny using the R packages meta and metafor (https://ubidi.shinyapps.io/easymeta/). All data were grouped by lineages and ethnicities using the random effects model.

## Results

Differences between cases and controls regarding various features are shown in Table 1. Birthplace (Central-East region, $p \leq 0.01$), and higher education levels were significantly more frequent in cases than in controls. Family history of PrCa ($p \leq 0.0001$) and breast cancer ($p = 0.09$) were more relevant inside cases. Regarding the Gleason score, 75% of cases at diagnosis were classified as PrCa moderated and poorly differentiated.

## The most frequent lineages and their geographic distribution

Paternal origins were traced back in 524 samples (152 cases and 372 controls), revealing a remarkable diversity within the haplotypes (Table 2). Twenty different haplogroups were

**Table 1. Characteristics of prostate cancer, and its matched controls randomly selected at a sample collection.**

| Characteristics | Cases | Controls | OR[a] | 95%CI | p-value |
|---|---|---|---|---|---|
| | n = 149 | n = 374 | | | |
| Age (Mean ± SD) | 64.70 ± 9.10 | 66.10 ± 8.90 | 0.98 | 0.96–1.00 | 0.10 |
| Birthplace [b] | | | | | |
| Mexico City | 88 (59.10) | 240 (64.20) | 1.00 | - | - |
| South | 10 (6.70) | 35 (9.40) | 0.86 | 0.40–1.82 | 0.69 |
| Central-West | 8 (5.40) | 29 (7.70) | 0.79 | 0.34–1.79 | 0.57 |
| Central-East | 35 (23.50) | 49 (13.10) | 2.13 | 1.28–3.54 | **<0.01** |
| North | 1 (0.70) | 11 (2.90) | 0.26 | 0.03–2.10 | 0.21 |
| East | 7 (4.70) | 10 (2.70) | 2.01 | 0.74–5.49 | 0.18 |
| Education | | | | | |
| Elementary school | 66 (44.30) | 170 (45.40) | 1.00 | - | - |
| Middle school | 22 (14.80) | 89 (23.80) | 0.59 | 0.34–1.03 | 0.06 |
| High school | 31 (20.80) | 74 (19.80) | 1.00 | 0.60–1.67 | 0.99 |
| University | 30 (20.10) | 41 (11) | 1.78 | 1.02–3.10 | **0.04** |
| Marital Status | | | | | |
| United vs. No united | 113 (75.80) | 295 (78.90) | 0.83 | 0.53–1.31 | 0.43 |
| Family history of cancer | | | | | |
| Prostate (Yes *vs*. No) | 17 (11.40) | 10 (2.70) | 4.70 | 2.08–10.5 | **≤0.0001** |
| Breast (Yes *vs*. No) | 7 (4.70) | 7 (1.90) | 2.52 | 0.86–7.33 | 0.09 |
| Histological differentiation [c] | | | | | |
| Well | 37 (24.8) | - | | | |
| Moderate | 60 (40.3) | - | | | |
| Poorly | 52 (34.9) | - | | | |

**Note:** OR, odds ratio; CI, confident intervals; SD, standard deviation; vs, versus

[a]OR adjusted by age at interview

[b]Birthplace: Mexico City (reference); Central-East: Hidalgo, Estado de México, Morelos, Puebla, Querétaro, and Tlaxcala; Central-West: Aguascalientes, Colima, Guanajuato, Jalisco, and Michoacán; East: Veracruz and Tabasco; North: Chihuahua, Coahuila, Nayarit, San Luis Potosí, Zacatecas, and Sinaloa; South: Chiapas, Guerrero, Oaxaca, and Yucatán

[c] Based on Gleason score at diagnosis. Bold values show significant p-values.

found; the Native American lineage -Q- was the most frequent (0.43), followed by R1b (0.29) and E (0.09) lineages (Fig 1; S1 Table). It is worthwhile to note the high Q proportions found, which depict the genetic features of the men from CDMX included in the present study. The Central Valley of Mexico (CVM) has been the homeland of several ancient civilisations, even before the Aztecs arrived [8]. This region is the primary destination for contemporary NAM migrants, as the city offers increased development opportunities. The last census of population and housing revealed that more than 289,139 NAMs are settled in Mexico City, representing 2.45% of seven million indigenous people living in Mexico [34].

E1b1a and E1a1b lineages were found in all regions except in those samples from the northern states, whereas Q and R1b were widely dispersed, exhibiting clinal distributions. In this setting, Q was most frequent in the southern region, exhibiting a decreasing trend northward, and R1b was most frequent in the central-western region, showing a decreasing trend southward (Fig 1).

## Ancestral burden contribution to the prostate cancer risk

Frequency differences between cases and controls were found in the haplogroups R1a, R1b, E (E1b1a/E1b1b), F-M89 (paragroup F(xGHIJK), represented by the lineages G, H, I, and J), and

**Table 2. Diversity parameters for non-recombinant Y-chromosome haplogroups of prostate cancer cases and its matched controls.**

| Haplogroup | N | h | Haplotype diversity | MPD |
|---|---|---|---|---|
| E1b1a | 7 | 7 | 1 ± 0.08 | 6.76 ± 3.63 |
| E1b1b | 40 | 40 | 1 ± 0.01 | 6.80 ± 3.27 |
| G2a | 11 | 11 | 1 ± 0.04 | 8.60 ± 4.31 |
| H | 3 | 3 | 1 ± 0.27 | 6.00 ± 3.92 |
| I1 | 6 | 6 | 1 ± 0.10 | 4.07 ± 2.36 |
| I2a1 | 9 | 9 | 1 ± 0.05 | 7.67 ± 3.95 |
| I2b1 | 7 | 7 | 1 ± 0.08 | 6.52 ± 3.51 |
| I2bx(I2b1) | 1 | 1 | 1 | 0 |
| J1 | 14 | 14 | 1 ± 0.03 | 5.15 ± 2.66 |
| J2a1 | 4 | 4 | 1 ± 0.18 | 6.83 ± 4.07 |
| J2a1b | 4 | 4 | 1 ± 0.18 | 4.83 ± 2.98 |
| J2a1h | 7 | 7 | 1 ± 0.08 | 4.33 ± 2.44 |
| J2a1xJ2a1-bh | 6 | 6 | 1 ± 0.10 | 6.80 ± 3.73 |
| J2b | 3 | 3 | 1 ± 0.27 | 6.33 ± 4.13 |
| L | 6 | 6 | 1 ± 0.10 | 10.33 ± 5.50 |
| O1b1 | 1 | 1 | 1 | 0 |
| Q | 224 | 224 | 1 ± 0.00 | 7.62 ± 3.57 |
| R1a | 6 | 6 | 1 ± 0.10 | 5.87 ± 3.26 |
| R1b | 153 | 153 | 1 ± 0.01 | 6.63 ± 3.15 |
| T | 11 | 11 | 1 ± 0.04 | 8.27 ± 4.16 |

**Note:** *N*, number of samples; *h*, number of different haplotypes; MPD, mean pairwise differences.

K-M9 (represented herein by L and T haplogroups). All subsequent analyses were done only in those haplogroups exhibiting significant values (Table 3).

**R1a contributed to the late-onset non-aggressive prostate cancer risk.** The R1a lineage revealed a contribution to PrCa risk (odds ratio, OR:8.04; 95% confidence intervals, $_{95\%}$CI:1.41–45.8), even after adjusting for age at diagnosis and family history (Table 3). Such risk was mainly related to late-onset PrCa (OR:7.34, $_{95\%}$CI:1.28–42.2) (Table 4) and histologically well-differentiated neoplasia (Gleason < 7; OR:23.3, $_{95\%}$CI:2.83–192.6) (Table 5).

A meta-analysis was conducted to support our findings (S1–S3 Figs). Overall, R1a showed a marginal contribution to PrCa development (OR:1.06; $_{95\%}$CI:0.99–1.14). Similar results were observed, mainly in European and Ashkenazi Jewish (OR:1.06; $_{95\%}$CI:1.01–1.12), followed by the African Americans (OR:1.15; $_{95\%}$CI:0.78–1.68). Of note was the contribution found in Latins (OR:5.45; $_{95\%}$CI:1.33–22.31). Nonetheless, a high heterogeneity value ($I^2 = 81\%$) was observed due to the small sample size (S1 Fig).

**E1b1a/E1b1b contributed to late-onset non-aggressive neoplasia.** Akin to R1a, E1b1a/E1b1b was associated with the late onset PrCa; OR:2.15, $_{95\%}$CI:1.02–4.53), and the histologically well-differentiated (Gleason < 7) neoplasia (OR:4.17, $_{95\%}$CI:1.43–12.1) (Tables 3 and 4). The meta-analysis results revealed a nuanced contribution of E1b1a/E1b1b lineages to PrCa risk (OR:1.15, $_{95\%}$CI:1.00–1.33). Variants within the E lineage, such as E1b1b1a1, have shown a marginal association with PrCa in European, and Ashkenazi Jews (S2 Fig). Despite the significant contribution of the Japanese men (OR:2.17, $_{95\%}$CI:1.16–4.06), it should be interpreted cautiously considering that such contribution was associated with the D haplogroup and other untagged ones. Regarding the PrCa's aggressiveness (Gleason < 7), E1b1a/E1b1b did not depict any contribution except for the data obtained in this study (Table 5).

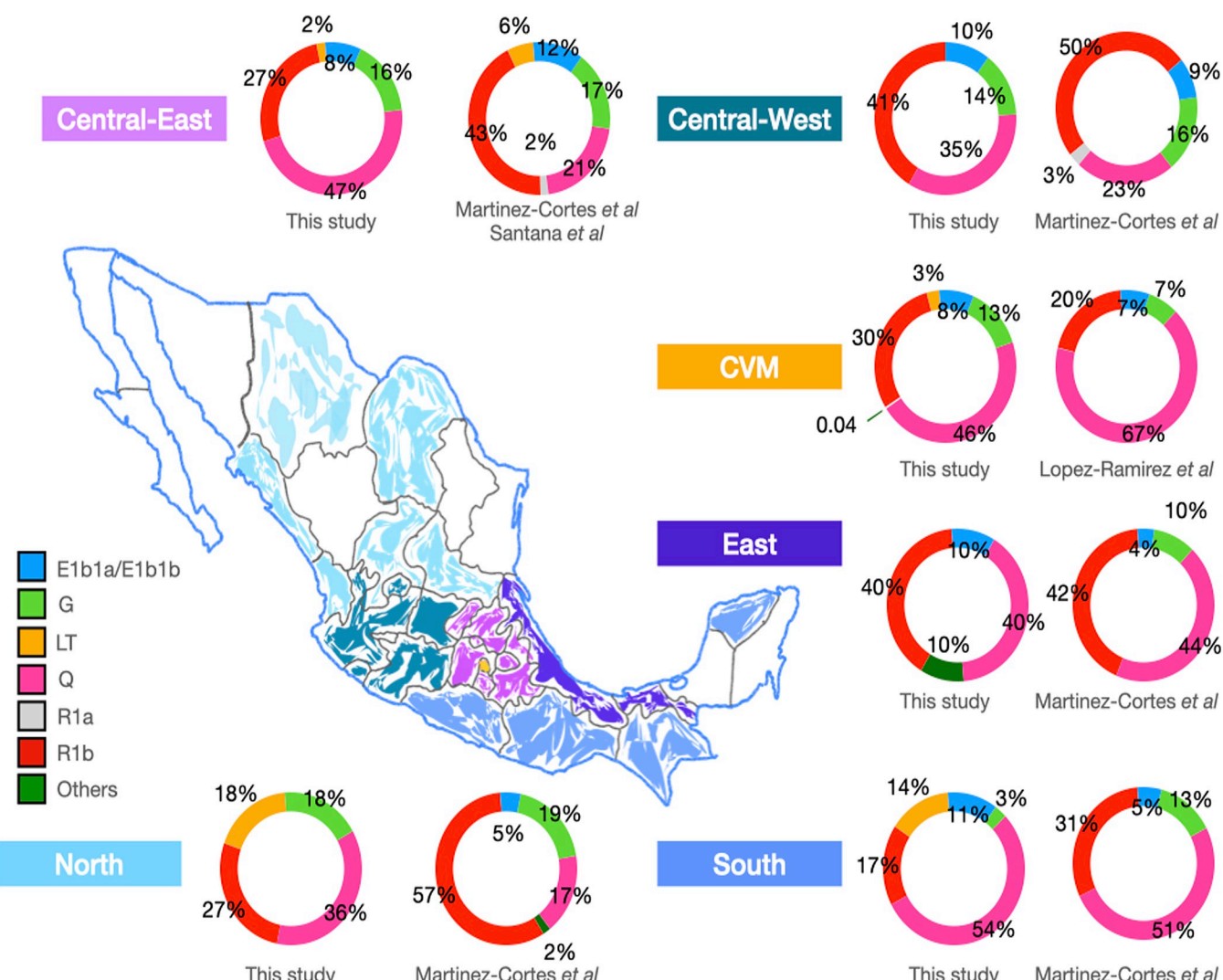

**Fig 1. Map of the Mexican Republic depicting the frequency distributions of the haplogroups found in the present study distributed by birthplace and its comparison with previous studies. Note:** The Central-East region comprising the states of Hidalgo, Estado de México, Morelos, Puebla, Querétaro, and Tlaxcala; the Central-West including the states of Aguascalientes, Colima, Guanajuato, Jalisco, and Michoacán; and the Central Valley of Mexico (CVM) was represented by Mexico City. The East region included the states of Tabasco and Veracruz. Regarding the North region it involved the states of Chihuahua, Coahuila, Nayarit, San Luis Potosí, Sinaloa, and Zacatecas whereas the South one included the states of Chiapas, Guerrero, Oaxaca, and Yucatán. The frequencies of population data were obtained from published reports: Lopez-Ramirez et al (2020); Martinez-Cortes et al (2012), and Santana et al (2014). Nonetheless, not all the states included by region were reported in these articles.

**Haplogroups G, H, I, and J contributed to non-aggressive neoplasia.** The combination of the haplogroups GHIJ considerably contributed to well-differentiated neoplasia (OR:4.01, $_{95\%}$CI:1.55–10.4) (Table 5). This contribution was marginal in the meta-analysis (OR:1.06, $_{95\%}$CI:0.98–1.15) (S3 Fig). Sub-lineages within the haplogroup I (i.e., I-M170 and I1c) have exhibited an evident contribution to PrCa in European and Ashkenazi populations. Overall, in Europeans and Ashkenazi Jews, the set of GHIJ haplogroups depicted a nuanced contribution (OR:1.06, $_{95\%}$CI:1.01–1.12), whereas in the northern Europeans, such contribution was marginal (OR:1.33, $_{95\%}$CI:0.91–1.94). Given the subtle frequency of the haplogroup H ($< 0.01$), it was excluded from the rest of the analyses.

**Table 3. Crude and adjusted Y-chromosome lineages contribution to prostate cancer development.**

| Y-chromosome lineages | Cases | Controls | OR[a] | 95%CI | OR[b] | 95%CI |
|---|---|---|---|---|---|---|
| | n = 149 (%) | n = 374 (%) | | | | |
| Q | 55 (36.90) | 170 (45.40) | 1 | | 1 | |
| R1b | 44 (29.50) | 109 (29.10) | 1.27 | 0.80–2.01 | 1.26 | 0.80–2.01 |
| R1a | 4 (2.70) | 2 (0.50) | **7.40** | **1.30–42.1** | **8.04** | **1.41–45.80** |
| E1b1a/E1b1b | 16 (10.70) | 31 (8.30) | 1.66 | 0.84–3.27 | 1.62 | 0.81–3.25 |
| GHIJ | 28 (18.80) | 47 (12.60) | **1.76** | **1.00–3.08** | 1.69 | 0.95–3.00 |
| LT | 2 (1.30) | 15 (4) | 0.40 | 0.09–1.83 | 0.44 | 0.10–1.99 |

**Note:** OR, odds ratio; CI, confident intervals

[a]OR adjusted by age at diagnostic or at interview

[b]OR adjusted by age at diagnostic or at interview, and family history of prostate cancer. Bold values show significant *p*-values.

## Comparisons with other populations

In those haplogroups contributing to PrCa, linear discriminant analyses were conducted. More than 9000 haplotypes were used for comparisons. In these analyses, the populations of interest were compared with those populations that could contribute to the genetic architecture of Mexican Mestizos and the Iberian Peninsula (Fig 2; S2–S7 Tables). In addition, the case-control haplotypes were compared with other Mexican carriers of the interest lineages from various geographic regions, including the CVM, Guanajuato (GTO), and Nuevo Leon (NL; S8 Table).

Predictably, both cases and controls exhibited a significant component from several regions that conform to the IBP, particularly with the region of Cataluña (CAT). It is noteworthy the groupings with Anatolian Greeks (ANG), Chuetas (CHU), Corsicans (COR), Cypriots (CYP), Italians (ITA), Jewish, Lebanese (LBN), Libyans (LBY), and Turkish and Cypriots (T&C). About the lineage E1b1a, some connections were found with LBN and LBY haplotypes. Similar connections were found in E1b1b, besides those connections with CYP. The cases and controls carriers of lineages G and I exhibited connections with CHU, COR, ITA, and T&C. In the case-control populations, carriers of haplogroup J showed a particular similarity to CHU and COR. Of note was the remarkable grouping between cases (50% of them) and Jewish populations (especially with Levites -LEVI-) regarding the lineage R1a. Controls presented a high probability of belonging to the IBP clades from the Madrid Community (MAD) and Aragon (ARA).

**Table 4. Adjusted Y-chromosome lineages contribution and early- or late-onset to prostate cancer development.**

| Y-chromosome lineage | Controls n = 374 (%) | Early-onset PrCa | | | Late-onset PrCa | | |
|---|---|---|---|---|---|---|---|
| | | Cases | OR[a] | 95%CI | Cases | OR[a] | 95%CI |
| | | n = 49 (%) | | | n = 100 (%) | | |
| Q | 170 (45.50) | 19 (38.80) | 1.00 | | 36 (36) | 1.0 | |
| R1b | 109 (29.10) | 14 (28.60) | 1.42 | 0.62–3.23 | 30 (30) | 1.23 | 0.70–2.15 |
| R1a | 2 (0.50) | – | – | – | 4 (4) | **7.34** | **1.28–42.2** |
| E1b1a/E1b1b | 31 (8.30) | 2 (4.10) | 0.40 | 0.06–2.50 | 14 (14) | **2.15** | **1.02–4.53** |
| GHIJ | 47 (12.60) | 12 (24.50) | 1.74 | 0.70–4.33 | 16 (16) | 1.64 | 0.81–3.32 |
| LT | 15 (4) | 2 (4.10) | 1.06 | 0.17–6.62 | 0 | - | - |

**Note:** OR, odds ratio; CI, confident intervals; PrCa, prostate cancer

[a]OR adjusted by age at diagnostic or at interview, and family history of prostate cancer. Bold values show significant *p*-values.

**Table 5. Adjusted Y-chromosome lineages contribution and the histological differentiation to prostate cancer development.**

| Y-chromosome lineages | Controls n = 374 (%) | Gleason <7 | | | Gleason ≥ 7 | | |
|---|---|---|---|---|---|---|---|
| | | Cases | OR[a] | 95%CI | Cases | OR[a] | 95%CI |
| | | n = 37 (%) | | | n = 112 (%) | | |
| Q | 170 (45.40) | 9 (24.30) | 1 | | 46 (41.10) | 1 | |
| R1b | 109 (29.10) | 7 (18.90) | 1.24 | 0.45–3.45 | 37 (33) | 1.25 | 0.76–2.08 |
| R1a | 2 (0.50) | 2 (5.40) | **23.3** | **2.83–192.6** | 2 (1.80) | 4.8 | 0.65–35.7 |
| E1b1a/E1b1b | 31 (8.30) | 7 (18.90) | **4.17** | **1.43–12.1** | 9 (8) | 1.19 | 0.52–2.71 |
| GHIJ | 47 (12.60) | 11 (29.70) | **4.01** | **1.55–10.4** | 17 (15.20) | 1.21 | 0.62–2.35 |
| LT | 15 (4) | 1 (2.70) | 1.28 | 0.15–10.8 | 1 (0.90) | 0.27 | 0.03–2.11 |

**Note:** OR, odds ratio; CI, confident intervals; PrCa, prostate cancer

[a]OR adjusted by age at diagnostic or at interview, and family history of prostate cancer. Bold values show significant $p$-values.

Regarding the regional contribution from other Mexican Mestizo populations, all case-control lineages exhibited a high clustering proportion with the NL population in northern Mexico. In connection with these findings, NL's haplotypes were compared to other populations (Fig 3). E1b1a, G, and R1a carriers showed high probabilities of being related to the Middle East (MEA) populations (average of 0.394 and 0.614, respectively). By contrast, E1b1b (0.310), I (0.735), and J (0.650) lineages presented their major average probabilities of clustering with IBP populations.

## Searching for a founder connection with prostate cancer

The case-controls networks depicted the contribution of each haplogroup to PrCa's risk (S4–S6 Figs). Overall, haplogroups E1b1a and E1b1b were more frequent amongst controls than cases. By contrast, haplogroups such as G2a, I2b1, J2a1h, J2b, and R1a were more frequent among the cases compared to the distribution of the controls.

To establish some connection between the haplotypes of cases and controls and their ancestral origins, those haplotypes contributing to PrCa were compared to various databases. More than 6,000 haplotypes whose haplogroup was determined with 14 Y-STRs were used for several comparisons. Such haplotypes were related to the populations that historically contributed to forging the genetic architecture of Mexican and IBP populations (Figs 4–9). Moreover, $R_{ST}$ analyses were also conducted to determine significant differences among populations (S9–S19 Tables).

Most of the haplotypes were related to various regions from the IBP, such as Andalusia (AND), ARA, Basque Country (PVA), CAT, Cantabria (CANT), Castilla y Leon (CYL), Valenciana Community (CVA), Galicia (GAL), La Rioja (RIO), MAD, and Navarra (NAV), among others. Some haplotypes connected with ANG, CHU (from Majorca), Jewish populations, and T&C, reinforce the findings obtained with the linear discriminant analysis.

Generally, cases and controls bearers of E1b1a haplogroups were connected with Arabian Gulf (ARAG), CYP, LBN, and Uganda (UGA) haplotypes. E1b1b cases and controls bearers presented a reasonable connection with those haplotypes from the CAT region and LBN (Fig 7).

Given that some databases included a minor number of YSTRs, MDSs with 6 YSTRs were also made to increase the number of populations for comparison (S7–S11 Figs; S11, S13, S15, S17 and S19 Tables). Of note was the relationship between some case-controls E1b1b carriers within the Jewish populations using 6-YSTRs, especially in those haplotypes that belonged to Ashkenazi (ASH), Sephardic (SEPH), and LEVI (S7 Fig). GIJ and R1a lineages, did not exhibit

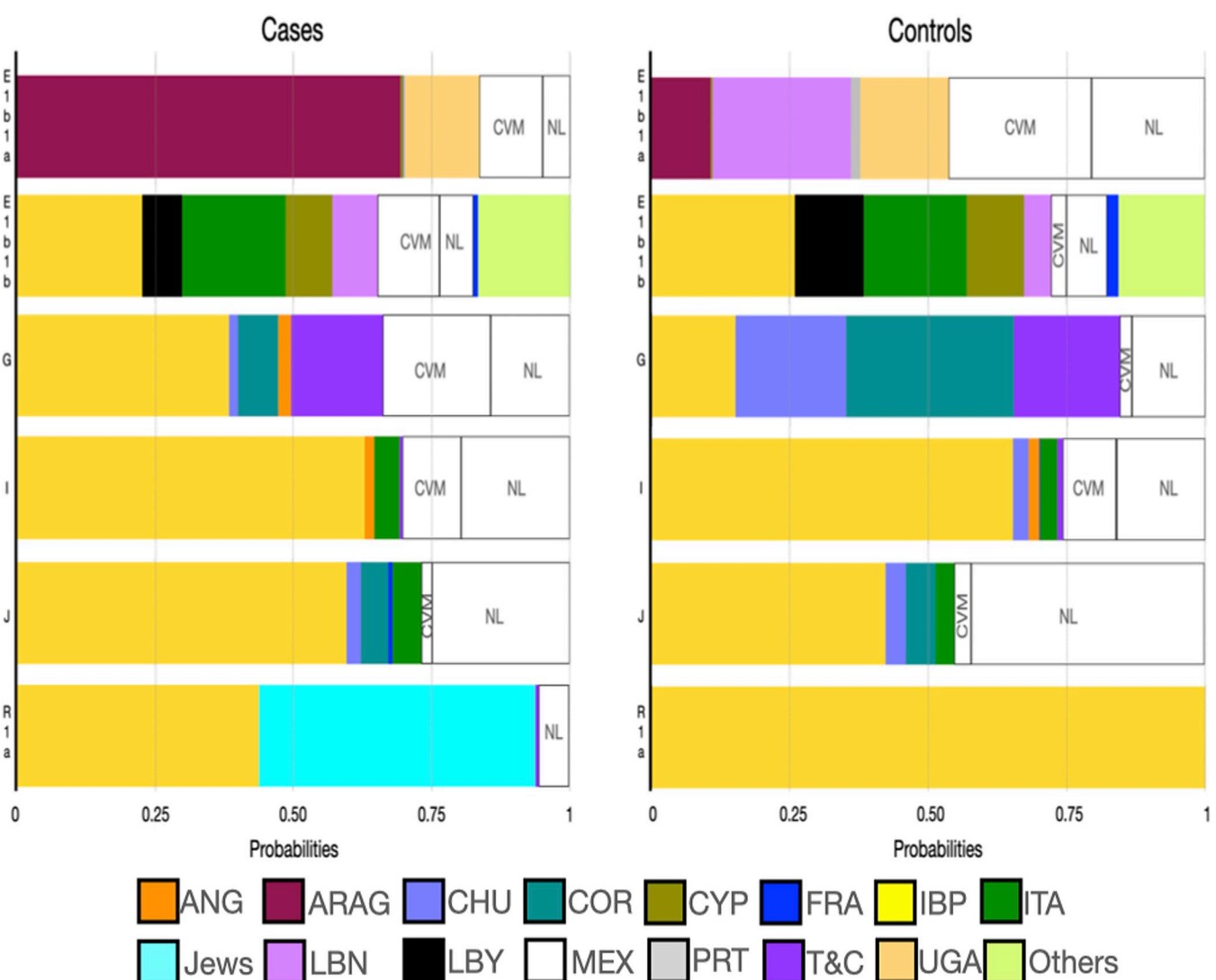

**Fig 2. Mean influence of the ancestral burden of the Y-chromosome lineages that contributed to prostate cancer obtained from linear discriminate analyses. Note:** The proportions included regions within the Mexican Republic, the Iberian Peninsula and those ancestral populations that contributed to their genetic architecture such as ANG, Anatolian Greeks; ARAG, Arabian Gulf, CHU, Chuetas; COR, Corsica; CVM, Central Valley of Mexico; CYP, Cyprus, FRA, France; IBP, Iberian Peninsula; ITA, Italy; LBN, Lebanon; LBY, Libya; MEX, Mexico; NL, Nuevo Leon; PRT, Portugal; T&C, Turkish and Cypriots; UGA, Uganda.

any significant difference with ANG, CHU, COR, CYP, ITA, LBN, LBY, SEPH, and T&C (S8–S11 Figs). Again, cases and controls were related to MEA populations. In addition to the previously mentioned populations, a connection with ASH Jews was found in the lineages I, J, and R1a. Notably, in this last lineage -R1a-, a remarkable relationship with IBP -Cohen- and SEPH-Jews was found with 14-YSTRs (Fig 9). Given the stress-values, these data should be interpreted with caution.

**The Native American legacy.** As mentioned before, the NAM lineages were the most frequent within the case-control populations. The Bayesian methods used to assign sublineages within the Q haplogroup suggested the presence of three sublineages: Q-MEH2 (Q1a), Q-L54 (Q1b1a2), and Q-M3 (Q1b1a1a). Q-MEH2 was related to *Chukchi* and *Tlingit* populations from North America and *Nahua* populations from Mesoamerica (S12 Fig). Q-L54 depicted a

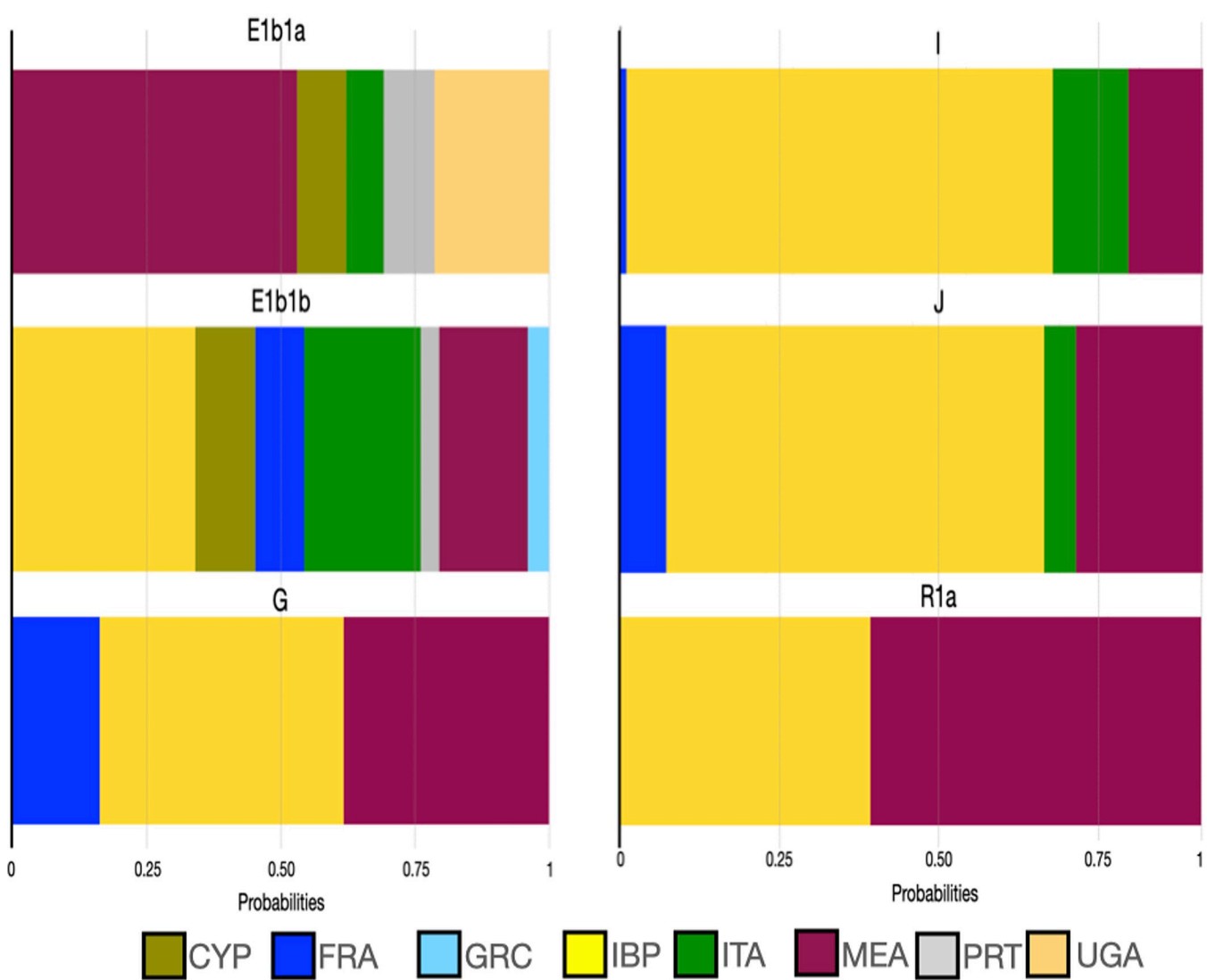

**Fig 3. Mean influence of the ancestral burden of the Y-chromosome lineages that contributed to prostate cancer obtained from linear discriminate analyses in samples from the Nuevo Leon region. Note:** The proportions included the Iberian Peninsula and those ancestral populations that contributed to its genetic architecture such as CYP, Cyprus, FRA, France; IBP, Iberian Peninsula; GRC, Greece; IBP, Iberian Peninsula; ITA, Italy; MEA, Middle East; PRT, Portugal; UGA, Uganda.

relationship with the NAM populations from the CVM, such as *Hñähñú*, *Nahua*, *Tepehua*, *Totonaca*, and *Úza*, as well as with haplotypes from Altai-Kizhi and the Tuva Republic (S13 Fig). The baroque network derived from Q-M3 demonstrated that this sublineage was the most frequent in the Mexican population and the least resolved phylogenetically (S14 Fig). The cases and controls carrying this lineage were related to *Hñähñú*, *Nahua*, *Popoloca*, *Tepehua*, *Totonaca*, and *Úza* contemporary NAMs populations.

## Discussion

Prostate cancer is the second most common cancer among men worldwide [1, 3]. The Y-chromosome has been considered a prototype biomarker as far as the health of males is concerned. Aberrant expression in several genes within this chromosome and alterations in the number of

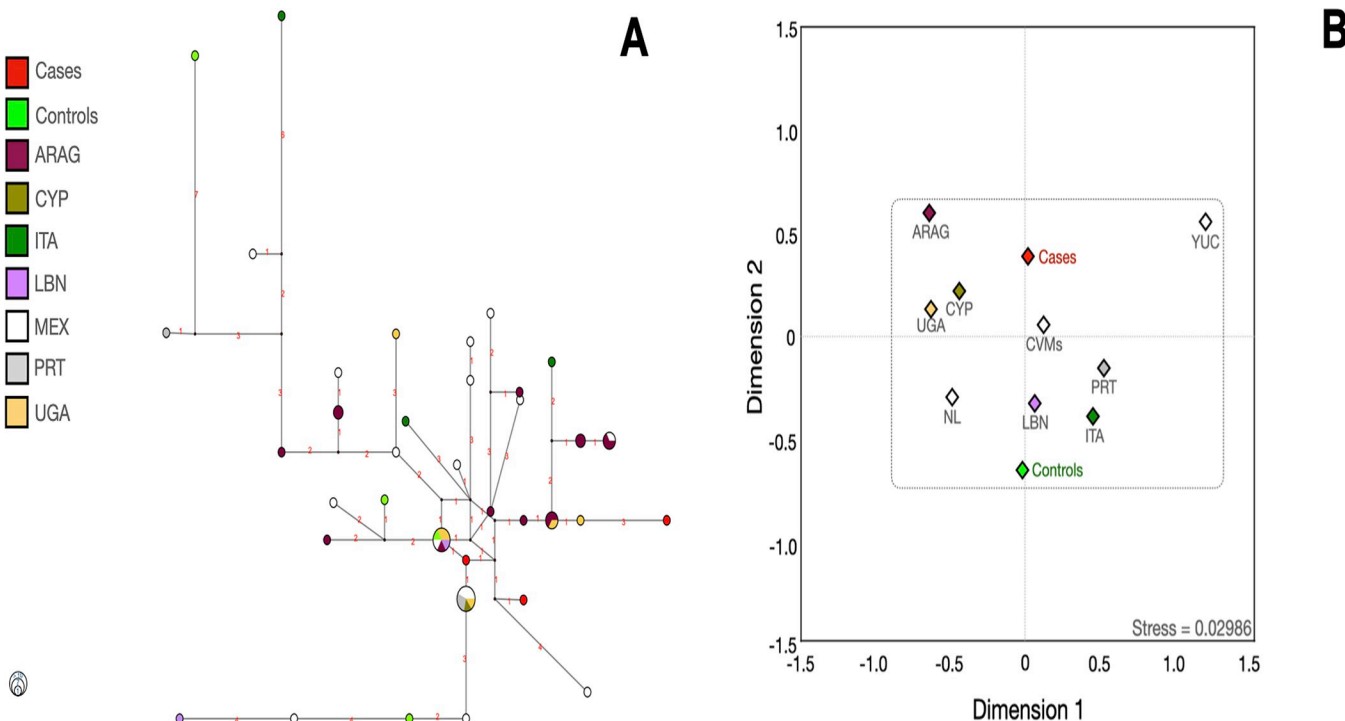

**Fig 4.** Median-joining network (**A**), and multidimensional scale plot of $R_{ST}$ values (**B**) of the E1b1a lineage with 14 Y-STRs using ethnicity as a criterion. **Note:** ARAG, Arabian Gulf; CVMs (Central Valley of Mexico, data from Santana *et al*); CYP, Cyprus; ITA, Italy; LBN, Lebanon; MEX, Mexico; NL, Nuevo Leon; PRT, Portugal; UGA, Uganda; YUC, Yucatan. Numbers in red represent the number of differences between one haplotype and other(s). All *p* values were adjusted with the method of false discovery rates. The dotted lines indicate that no significant differences were found among the populations. The circle size indicates the approximate number of individuals with shared haplotypes; the smallest circles represent one individual.

tandem repeats have been related to PrCa's development and progression [5, 35]. Among the risk factors, ethnicity has been considered well-established [25, 36]. Such findings keep the effort alive to continue its study at different levels and evaluate its contribution to male-specific disorders such as PrCa. Herein, the paternal origin of 152 PrCa cases and 372 controls from the Mexican Mestizo population was traced back to explore the patrilineal lineages' contribution to the PrCa's risk. Overall, the paternal ancestry distributions were similar to those reported in prior Mexican studies, with nuanced differences related to the geographic region [37].

### Haplogroups contributing to prostate cancer

**A medical context.** The contribution of E1b1a/E1b1b, G, I, J, and R1a to non-aggressive late-onset PrCa development was revealed in the present study. Such findings were consistent with prior reports on haplogroups such as E (Asian and Latins), I (Ashkenazi Jewish, Europeans and Latins), and R1a (Ashkenazi and Europeans) [38–41]. The Y-chromosome is carrier of onco- and tumour suppressor genes such as the testis-specific protein Y-encoded (*TSPY*), which has shown an increased expression in PrCa and other malignancies [42]. The interaction between *TSPY* and the eucaryotic elongation factor 1 alpha has demonstrated its role in cell proliferation (*in vitro*) and tumorigenesis (*in vivo*) [43, 44]. In addition, the association between the Y-haplogroups and PrCa development could be explained by the presence of microdeletions in different ancestral backgrounds [45]. Modifications in *TSPY* (profits and

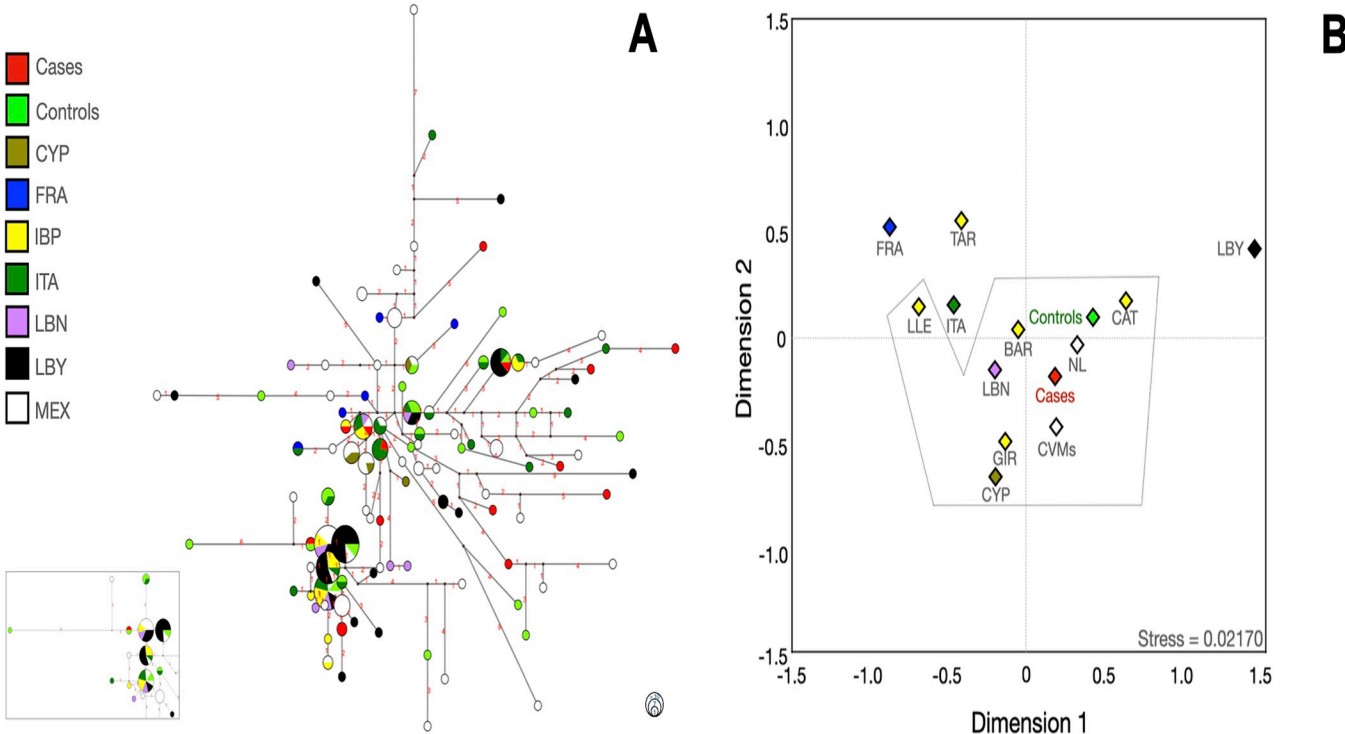

**Fig 5.** Median-joining network (**A**), multidimensional scale plot of $R_{ST}$ values (**B**) of the E1b1b lineage with 14 Y-STR using ethnicity as a criterion. **Note:** BAR, Barcelona; CAT, Cataluña; CVMs (Central Valley of Mexico, data from Santana *et al*); CYP, Cyprus; FRA, France; GIR, Girona; IBP, Iberian Peninsula; ITA, Italy; LBN, Lebanon; LBY, Libya; LLE, Lleida; MEX, Mexico; NL, Nuevo Leon; TAR, Tarragona. Numbers in red represent the number of differences between one haplotype and other(s). All *p* values were adjusted with the method of false discovery rates. The dotted lines indicate that no significant differences were found among the populations. The circle size indicates the approximate number of individuals with shared haplotypes; the smallest circles represent one individual.

losses) have shown frequency differences in European patrilineal backgrounds such as Italian, German and Spanish, among others [45].

Nonetheless, discordant findings have also been reported even in populations with similar ancestral burden [39–41]. These discrepancies might be explained by the typical genetic architecture of each population. Particularly, Latins are a young population with a recent and heterogeneous miscegenation [9]. One of the major causes of spurious genetic associations is the admixture [46]. Furthermore, the discrepant findings might be a result of the statistical simplification used in genetic studies with medical and population models. Such models assume random assortment and mating to simplify the statistical analyses [47]. Because of this simplification, the social and demographic dynamics -implicit in the population's genomes- might not be fully represented, skewing the results obtained from them [46]. From this perspective, it is probable that there is both under- and over-representation, influencing the findings supported in frequency distributions. Along with these two possibilities, besides the frequency of uniparental lineages, other segments in the autosomal genome could be affected [48]. High rates of intracommunity and consanguineous marriages have been documented in Ashkenazi Jews, in whom more than 90% of all pathogenic BRCA1/2 variants have been found [48, 49]. Thus, it is likely that these studies were reflecting the contribution of BRCA's variants more than the contribution of the lineages. As in previous studies, breast cancer and PrCa family history was critical in developing this latest pathology, confirming the positive correlations between these two types of neoplasia [50, 51]. Remarkably, the familial relative risk of PrCa

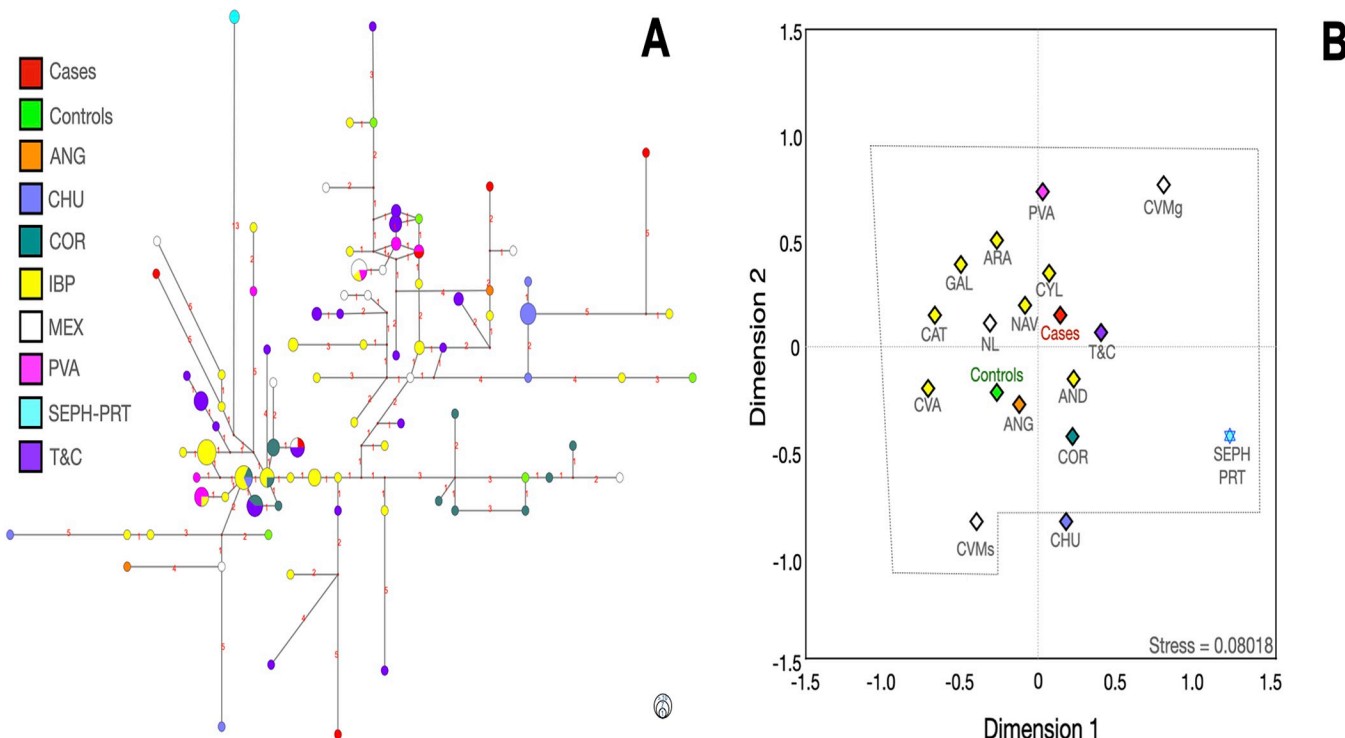

**Fig 6.** Median-joining network (**A**), multidimensional scale plot of $R_{ST}$ values (**B**) of the G lineage with 14 Y-STRs using ethnicity as a criterion. **Note:** AND, Andalusia; ANG, Anatolian Greeks; ARA, Aragon; CAT, Cataluña; CHU; Chuetas; COR, Corsica; CVA; Valencian Community; CVMg (Central Valley of Mexico, data from Gomez et al); CVMs (Central Valley of Mexico, data from Santana et al); CYL, Castilla and Leon; GAL, Galicia; IBP, Iberian Peninsula; MEX, Mexico; NAV, Navarra; NL, Nuevo Leon; PVA, Basque Country; SEPH-PRT, Sephardic Jews from Portugal; T&C, Turkish and Cypriots. Numbers in red represent the number of differences between one haplotype and other(s). All p values were adjusted with the method of false discovery rates. The dotted lines indicate that no significant differences were found among the populations. The circle size indicates the approximate number of individuals with shared haplotypes; the smallest circles represent one individual.

has shown contributions closer than 40%, replicated even in several ethnicities, reinforcing the genetic contribution in developing this ailment [36, 52].

A similar explanation could have occurred in the Mexican Mestizo population whose recent rise (~ 500 ya, fairly close than 20 generations), affects the inbreeding rates; remarkably high in this population [53]. Also, of note is the sex-biased admixture present in Latin populations, which could cause spurious associations. Thus, our results should not be taken categorically, as they could be reflecting the effects of several demographic events influencing the distributions of lineages in the studied population. Furthermore, the demographic processes could differ over time between males and females altering the uniparental markers' genetic frequencies and the medical studies derived from them [54].

On the other hand, the ancient human dispersal movements most probably provoked adaptive responses to a wide range of behaviours (i.e., climate, diets, presence of arsenic in the rivers' water, etc.), modifying the genetic diversity. It is likely that some haplogroups were more adapted to certain selective pressures and, in turn, were fixed in the populations. As a result of positive selection, fecundity could increase, while in other cases, with selective sweep, the fertility could decrease [45]. Sex-specific expression patterns have been reported both in sexual and autosome chromosomes with remarkable variation amongst populations and geographic regions [54]. These evolutionary theories might explain the differential lineage frequency depending on the geographic regions, and probably the presence of diseases such as PrCa which has been associated with infertility [45, 55].

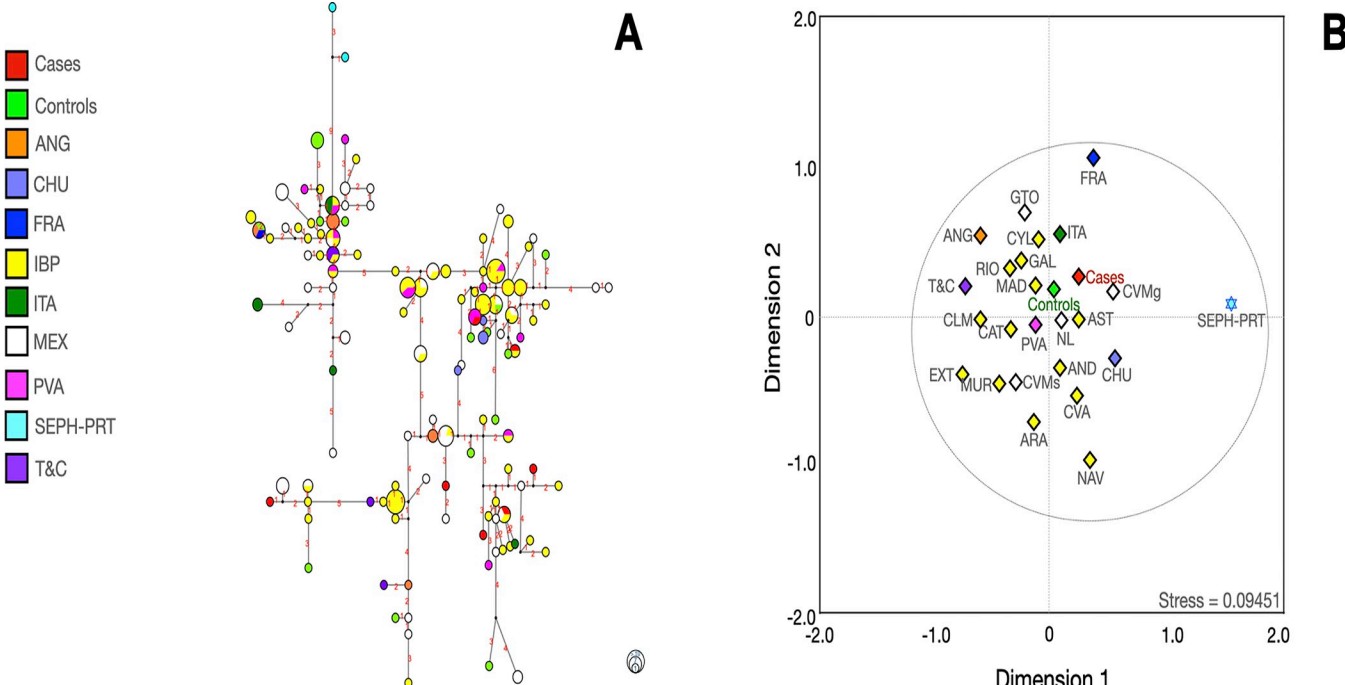

**Fig 7.** Median-joining network (**A**), multidimensional scale plot of $R_{ST}$ values (**B**) of the I lineage with 14 Y-STRs using ethnicity as a criterion. **Note:** AND, Andalusia; ANG, Anatolian Greeks; ARA, Aragon; AST, Asturian Community; CAT, Cataluña; CHU; Chuetas; CLM, Castilla la Mancha; CVA; Valencian Community; CVMg (Central Valley of Mexico, data from Gomez et al); CVMs (Central Valley of Mexico, data from Santana et al); CYL, Castilla and Leon; EXT, Extremadura; FRA, France; GAL, Galicia; GTO, Guanajuato; IBP, Iberian Peninsula; ITA, Italy; MAD, Madrid; MEX, Mexico; MUR, Murcia; NAV, Navarra; NL, Nuevo Leon; PVA, Basque Country; RIO, La Rioja; SEPH-PRT, Sephardic Jews from Portugal; T&C, Turkish and Cypriots. Numbers in red represent the number of differences between one haplotype and other(s). All p values were adjusted with the method of false discovery rates. The dotted lines indicate that no significant differences were found among the populations. The circle size indicates the approximate number of individuals with shared haplotypes; the smallest circles represent one individual.

Some of the possibilities mentioned earlier (i.e., inbreeding, founder effect, and the heterogenous admixture) could explain the differences found between cases (23.5%) and controls (13.1%) in the Central-Eastern region. This geographic region was represented by the political states of Hidalgo, Estado de Mexico, Morelos, Puebla, Queretaro, and Tlaxcala. Once *Conversos* arrived in *Nueva España*, they settled in present-day Mexico City. In 1552, mining concessions and exploitation were granted exclusively to the "Spanish", a nationality that camouflaged Semitic religions [8]. Then, the most important settlements of *Converso*s were documented in the mining cities of Estado de Mexico, Hidalgo, GTO, Guerrero, Puebla, Queretaro, Tlaxcala, and Zacatecas [9, 56]. Former reports have demonstrated the increased presence of non-Indigenous lineages in Hidalgo and Queretaro states [8, 57]. Hidalgo and Tlaxcala were also important reservoirs of Jewish migrations [8, 56]. In recent times, the economic development of CDMX has been a focal point for internal migratory movements. States such as Estado de Mexico, Hidalgo, Morelos, Puebla, and Tlaxcala have shown the highest rates of migration (36.6%) to CDMX, followed by GTO, Michoacan, Queretaro, San Luis Potosi, and Veracruz with 28.1% [58]. Nonetheless, such migratory rates are not homogeneous regarding the frequency of the lineages, which could explain the differences found in the mentioned geographic region; lineages such as GHIJ and R1a were more prevalent in the case group. By contrast, E1b1a and E1b1b were more frequent in controls.

Given that mining exploitation is carried out in the mentioned states, it is also likely that the contribution found was associated with environmental exposure. In this setting, high

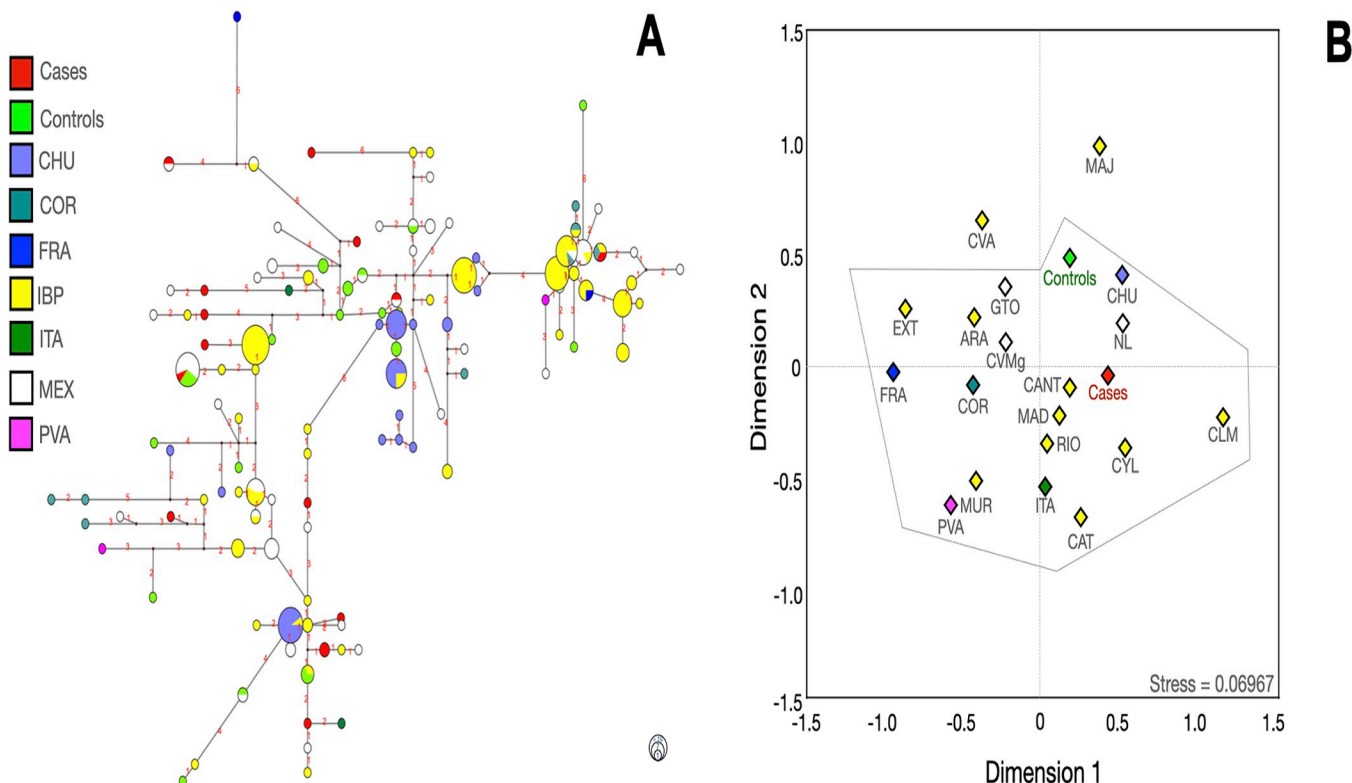

**Fig 8.** Median-joining network (**A**), multidimensional scale plot of $R_{ST}$ values (**B**) of the J lineage with 14 Y-STRs using ethnicity as a criterion. **Note:** ARA, Aragon; CANT, Cantabria; CAT, Cataluña; CHU; Chuetas; CLM, Castilla la Mancha; COR; Corsica; CVA; Valencian Community; CVMg (Central Valley of Mexico, data from Gomez et al); CYL, Castilla and Leon; EXT, Extremadura; FRA, France; GTO, Guanajuato; IBP, Iberian Peninsula; ITA, Italy; MAD, Madrid; MAJ, Majorca; MEX, Mexico; MUR, Murcia; NL, Nuevo Leon; PVA, Basque Country; RIO, La Rioja. Numbers in red represent the number of differences between one haplotype and other(s). All *p* values were adjusted with the method of false discovery rates. The dotted lines indicate that no significant differences were found among the populations. The circle size indicates the approximate number of individuals with shared haplotypes; the smallest circles represent one individual.

arsenic concentrations have been reported in the states of Hidalgo, Morelos, Puebla, and Zacatecas; the exposure to this metalloid has been associated with PrCa [59–61].

Of further note was the relationship of the R1a lineage to PrCa. Such a contribution was found in well-differentiated and late-onset PrCa. Despite this, it is essential to note that the number of cases was low, and in turn, the confidence intervals that support these findings were wide. Although it is probable that the low representation of this lineage could reflect a sampling error, in fact R1a is one of the least frequent lineages in the Mexican population [9]. The frequency of R1a in the several databases from the Mexican population represented by 2,388 unrelated individuals included in all analyses (0.0108) was similar to the frequency found in the present study (0.0114). In addition, the sample analysed was randomly selected and the genotype characterisation was blinded. Ultimately, the case distributions according to Gleason score at diagnosis was comparable to those observed at national level, and the expected for a country without a preventive program for early prostate cancer detection [62]. In turn, the selection bias and the measurement errors as possible explanations of the findings obtained might not be applicable.

**A historical context.** Regarding the geographic distribution of the haplogroups contributing to PrCa, these have been connected with European, Middle Eastern, and African ancestries [7, 63, 64]. Thus, the contribution of the PrCa lineages found herein is unsurprising and

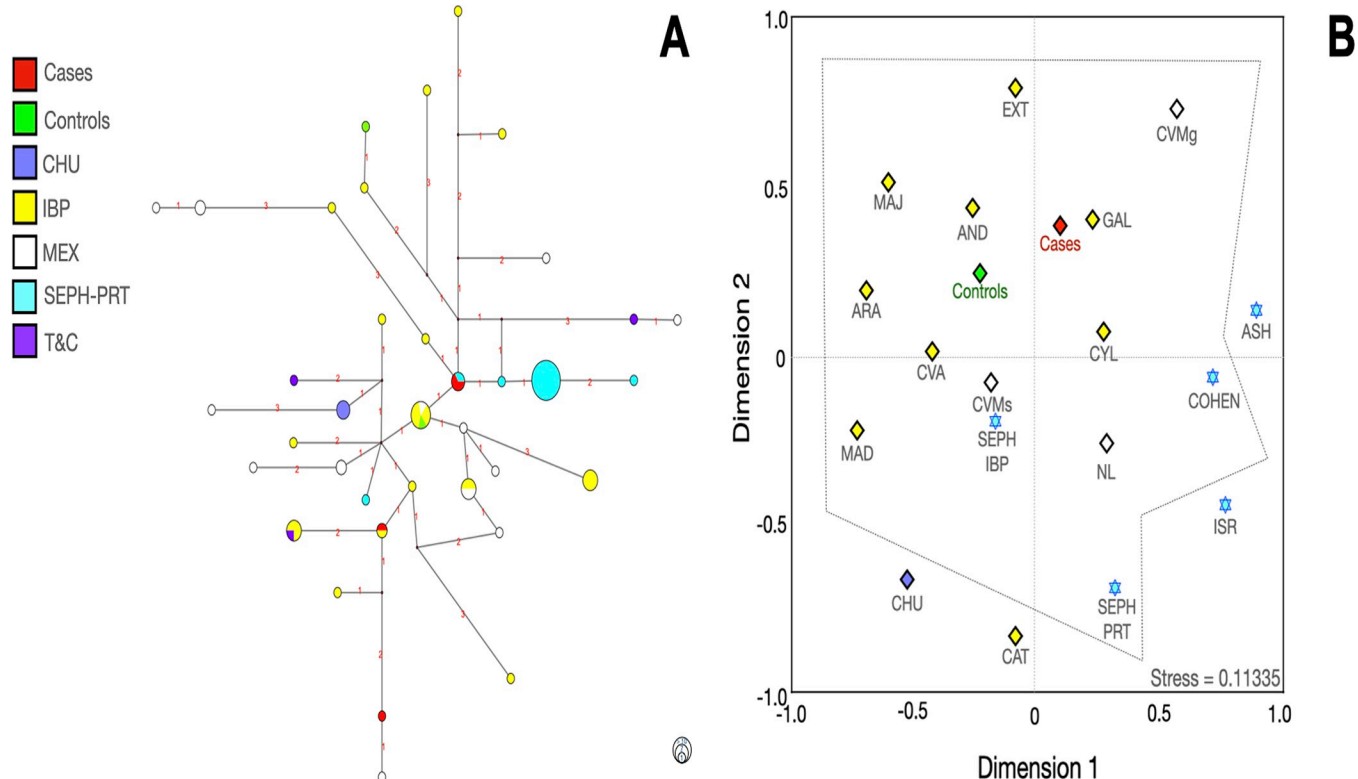

**Fig 9.** Median-joining network (**A**), multidimensional scale plot of $R_{ST}$ values (**B**) of the R1a lineage with 14 Y-STRs using ethnicity as a criterion. **Note:** AND, Andalusia; ARA, Aragon; ASH, Ashkenazi Jews; CAT, Cataluña; CHU; Chuetas; COHEN, Cohen Jews; CVA; Valencian Community; CVMg (Central Valley of Mexico, data from Gomez et al); CVMs (Central Valley of Mexico, data from Santana et al); CYL, Castilla and Leon; EXT, Extremadura; GAL, Galicia; IBP, Iberian Peninsula; ISR, Jews from Israel; MAD, Madrid; MAJ, Majorca; MEX, Mexico; NL, Nuevo Leon; SEPH-IBP, Sephardic Jews from the Iberian Peninsula; SEPH-PRT, Sephardic Jews from Portugal; T&C, Turkish and Cypriots. Numbers in red represent the number of differences between one haplotype and other (s). All *p* values were adjusted with the method of false discovery rates. The dotted lines indicate that no significant differences were found among the populations. The circle size indicates the approximate number of individuals with shared haplotypes; the smallest circles represent one individual.

reinforces the findings described in prior studies [38–41]. To our knowledge, the present study is the first report on Latin males. Of note, these lineages are not native of the Americas, arriving together with the Iberian heritage and its subsequent trans-Atlantic migrations [9].

**Lineage E: E1b1a & E1b1b.** Overall, haplogroup E appeared in Northeast Africa (around 40,000 years ago, ya) and then left the continent headed towards the Middle East [65]. The history of E1b1a in the Americas has been connected with West African populations, particularly with those who arrived in the New World through the slave trade brought by the Spanish colonists. Even though the origins of the enslaved Africans are heterogeneous, the historical reports allude to the Bight of Biafra (Cameroon, Equatorial Guinea, northern Gabon and eastern Nigeria) [66]. Present-day countries such as Cape Verde, Senegal, Guinea, Sierra Leone, *Côte D'Ivoire*, and Ghana also contributed to the African ancestry in the Americas [67]. Such countries exhibited PrCa incidence rates between 32.0 and 68.2 [68].

Nonetheless, the E1b1b haplogroup has been inaccurately related only to Africa; its history also goes back to North Africa and present-day Cyprus, Israel, Jordan, Lebanon, and Syria (the Levant region) [19, 20]. In these countries the PrCa incidence rates are variable, although Cyprus and Israel have exhibited the highest rates (between 32.0 to 44.6) [68].

**Lineages G, I, and J.** Haplogroup G arose in the eastern edge of the Middle East around 30,000 ya [65]. This haplogroup has been related to Circum-Mediterranean populations

(South Europe, Middle East, and North Africa), being mainly present in Moroccans (G-M201, ~0.330) and Turkish & Cypriots (G2a, ~0.160) [21, 69]. G-M201 could have also been introduced to the Iberian Peninsula during the Neolithic period [65, 70]. G haplogroup shares recent common ancestors (M168 and M89) with the haplogroups H, I, and J [65]. The haplogroup I belonged to the Middle East clan, which was subsequently spread into central Europe (~21,000–28,000 ya) [65]. This lineage has also been reported in North Africans (0.003), Turkish and Cypriot populations (< 0.001 for I1, and around 0.068 for I2), and in the Middle East (range: 0.030–0.080) populations [69–72]. In Europe, the haplogroup I has exhibited a wide variation in its frequency distribution. Crete, Greece, and Sardinia have reported remarkable frequencies of 0.100, 0.190, and 0.326, respectively [72]. Haplogroup J is the youngest (~15,000 ya), being the Fertile Crescent area in the region from its emergence [19, 65]. This region corresponds to the present-day countries of Egypt, Iraq, Israel, Jordan, Lebanon, Palestine, Syria and Türkiye [65]. The sublineage J1 has been associated with the Islamic conquerors and the Semitic-speaking populations, exhibiting high frequencies in the Middle East [19]. By contrast, the sublineage J2, the most widely distributed in Europe, has shown a particular distribution along the Mediterranean basin, which is also related to the Neolithic farmers [19, 70]. These two clades (J1 and J2) have shown an enormous diversity within Jewish populations [73]. Of these, the Southern European countries showed PrCa incidence rates ranging from 44.6 to 157.5 [68].

**Lineage R1a.** Haplogroup R1a could have originated in southern Russia (10,000–15,000 ya), although other authors have it assigned to the Middle East with its subsequent dispersal to Europe during the Palaeolithic age [65, 70, 74]. In Ashkenazi Jewish, R1a is considered one of the most frequent haplogroups, presenting a remarkable diversity [73, 74]. Carriers of the R1a clan, Jewish (Ashkenazi and Levites) and non-Jewish, shared one common male ancestor who lived fairly close to 1,743 years ago [74]. Cretans (~0.110), Greeks (~0.160), Italians (range: ~0.011–0.065), and Turkish (range: ≤0.0001–0.116) have also shown considerable frequencies [72].

## Circum-Mediterranean genetic contribution to prostate cancer and the patrilineal Mexican background

The intricate amalgamation, both historical and genetic, of the colonisers who settled down in the *Nueva España* (current Mexico) has illuminated: 1) the genetic connections between Mexican men, with Iberians and Middle Easterners and 2) the remarkable non-Native Y-chromosome diversity found amongst the Mexican Mestizos. These attributes were consistent with previous studies, reinforcing the patrilineal portrait produced by 500 years of admixture [9, 14, 75]. Nonetheless, clarifying the connections between cases the Mexican cases with Chuetas and the role of R1a is of paramount importance.

Chuetas are considered Crypto-Jewish descendants inhabiting the Balearic Islands since the fifth century AD [71]. These islands are located in the western Mediterranean Sea, with Majorca and Minorca as the main islands. As an ethnic-religious minority, the inbreeding practices could have been continuous, maintaining its original genetic background. The frequency reported to haplogroups G (~0.100), I (~0.040), and J (J1 = 0.180 and J2 = 0.330) are higher in Chuetas than the overall frequencies within the IBP (G = 0.050, I = 0.060, and J = 0.010) [16, 71]. Similar frequencies to those exhibited by Chuetas have been reported in Sephardim populations G (~0.160), I (~0.010), and J (J1~0.220, and J2~0.250) [71]. These last lineages (J1 and J2) have connected Chuetas with their Middle Eastern ancestry [70, 71]. Hence, the connection found between PrCa cases and Chuetas could mirror the *Converso* migrations to *Nueva España*. Enormous genetic similarities have been documented amongst

Jewish populations (Ashkenazim, Sephardim, and Middle Easterners), whose origins converge in the Middle East [76]. Although the Spanish ancestry among Europeans was the most prominent in Mexico, Portuguese migrations have also been documented [57]. Former studies have demonstrated the contribution of Sephardim Jewish to the Latin/Hispanic genomes [57, 77].

When "Decrees of Expulsion" was established (1492), Sephardim groups (from Portugal and Spain) tried to leave the Peninsula, moving to other regions within Europe, the Middle East, North Africa, and *Nueva España* [21, 56]. It is noteworthy that the discovery of the Americas (1492) matched coincided with the Jewish expulsion; at least 8,000 Jewish families set sail at the same time as Columbus to the New World [78]. Hence, *Conversos*, who were indeed Crypto-Jews and Crypto-Muslims, migrated to the Americas, participating in its discovery and colonisation [56]. During the mid-sixteenth century, Portuguese Jews migrated to the New World; Portuguese was considered synonymous with Jewish and *Conversos* [21, 68, 77]. In order to facilitate their movements through the Circum-Mediterranean regions, Jewish communities settled along the eastern coast of the Mediterranean Sea [56, 78]. During the *Reconquista* of the Christian forces, Muslims that inhabited Galicia and Cantabria were relocated to Andalusia; the last Muslim stronghold was Granada [9, 16, 19]. The Andalusian region maintained high proportions of Sephardic and Arab surnames that were adapted to the Christian religion due to the necessity to mimic non-Jewish populations [79]. In addition, the provinces of Lucena (located in Andalusia) and Castilla La Mancha (e.g., Castilla and Toledo, mainly) were also inhabited by Jews [56].

**Nuevo Leon as a reservoir of the Sephardic legacy?** Despite the NL state not being included among the analysed samples, the linear discriminant analysis found a remarkable representation of this state and its Middle Eastern proportions (~0.600). As mentioned before, the lineages that contributed to the PrCa risk were connected to the circum-Mediterranean populations that influenced the southeaster regions of Spain. At least 100 Sephardim families colonised the Northern kingdoms of NL and *Nueva Extremadura* (current state Coahuila) [78]. Such ancestral origins might explain the remarkable Middle Eastern proportions found in NL. E1b1a presented connections with the Arabian Gulf, Cyprus, Lebanon, and Uganda. The presence of this last country could represent the signature of the enslaved Africans in the Americas [67]. The other genetic connections might be associated with the southeaster region of the Iberian Peninsula, which has been extensively illuminated. E1b1b was related to Catalonia and Lebanese populations. Catalonia migrations were documented, particularly in the northeaster region of Mexico, mainly in the NL state [57]. E1b1b could also represent the bidirectional migrations between the IBP and North Africa and the Arabian gene flow through this land bridge [19].

Without significant differences, the genetic closeness among cases and NL, CHU, Sephardim, T&C, and CLM populations could be reinforced by the presence of the haplogroup R1a mainly amongst the cases. Of note, R1a carriers were primarily connected with the Levite Jewish haplogroups. R1a clade, with its particular sublineages, is one of the most common within the Levite and non-Levite contemporary Jewish populations (range: 0.079–0.146) [74]. Also, R1a has shown a five-fold higher presence in Sephardim (0.050) rather than in IBP populations (0.010); similar frequencies have been reported in Chuetas (0.040) [16, 71]. Levite Sephardim have been documented in Catalonia (Gerona) during Medieval Spain [9, 54]. In Mexico, the main reservoirs of R1a carriers have been reported in the States of NL (~0.011), and GTO (0.347) [15]. This last state was represented within the studied samples.

Our study also represented contemporary migrations from Yucatan to CDMX. South Mexican states, such as Campeche and Yucatan, were also occupied by *Conversos* because of the Jewish community's affinity to reside near ports and border cities for easy access to ships when needed. Furthermore, in these two states, it was relatively easy to avoid the certification of

Christian *pureza de sangre* (pure blood) [56]. Merida, a city in the Yucatan peninsula, has shown ASIR values to PrCa closer to thirty [1]. This value is below those ranges reported in Latin populations (47.52 to 62.29). An explanation of this figure could be associated with the highest Native American proportions (0.543, this study) in the southern part of the Mexican Republic, diminishing the ASIR values, as these are generally reported. The Asian ancestry presented the lowest PrCa risk [1]. Such ancestry was carried by the first colonists during the different migrations that contributed to the peopling of the Americas around 9,699 to 27,479 years before the present [8]. Lineage Q is among the most prominent in the Mexican population, exhibiting remarkable frequencies southward [8, 9].

**A Converso possible founder effect?** In the context of PrCa's risk, the odds ratios reported in the present study were related to those found in Hispanics (OR:3.20, $_{95\%}$CI:2.64–3.92), and Europeans (OR:3.80, $_{95\%}$CI:3.62–3.96) [64]. Nonetheless, due to the demographic events that influenced in the Iberian gene pool, setting apart the Middle Eastern legacy (Jewish and Muslim) from the Iberian heritage proves to be challenging. Sephardim genetic signatures have been demonstrated through identity-by-descent, autosomes, and Y-chromosome studies within Latin populations [76, 77]. Notably, the ancestries from the Iberian Peninsula have shown a remarkable contribution to PrCa, mirrored in the Globocan [68]. Nevertheless, considerable heterogeneity regarding the incidence of PrCa across the IBP regions has been documented. The average incidence of PrCa in Spain in the last three years (2021–2023) has been reported as 31.88; the Northwest (Galicia and Asturias) and the Southwest (Andalusia) regions have been identified as having the highest risk of PrCa [80–82]. In Portugal, the predicted incidence (2020) has been estimated at 116.60 [83]. The most remarkable ASIRs have been reported in the Azores islands (98.4), followed by the North (95) and the South regions (78.60) [83]. Interestingly, the Azores archipelago has presented a stronger Jewish influence, particularly through the J lineage [84]. Among Middle Eastern countries, it is noteworthy that Jews from Israel have shown the highest overall cancer rates [63]. Particularly in PrCa, the ASIR values reported in Israel (84.30), Lebanon (~74), Cyprus (71.30), and Türkiye (range: 35.10–47.40), which could reinforce a possible founder effect [7, 63, 85, 86]. E, I, J, and R1a have been associated with PrCa in Europeans and Ashkenazi Jews [39, 41].

A possible Jewish founder effect may be associated with certain variants within *BRCA* (i.e., c.185delAG, 5382insC, c5946del, c7579delG, c5159C>A, c9693delA) genes. These and other variants within these two genes have been related to breast cancer, worldwide [87, 88]. Some *BRCA1* and *BRCA2* variants have exhibited more frequencies in Jewish *versus* non-Jewish populations [89–92]. *BRCA1*-rs1799966 has shown twice the risk to PrCa (OR:2.30, $_{95\%}$CI:1.36–3.91) in Mexican Mestizo men [51]. The samples examined in the current study are the same as those explored previously. Nonetheless, certain controversies have been reported [91, 93, 94]. Even though multi-ethnic populations represent a melting pot of diversity to explore the founder effects through uniparental markers or markers depicting regional ancestral burden, the findings could be skewed by population heterogeneity in other genetic variants. Thus, the discrepancies found amongst different studies could reflect the genetic background of the population studied rather than the differences between cases and controls [46, 95]. Other Jewish founder mutations in *BLM* have been reported in Mexican Ashkenazi women [94].

## Other risk factors associated with prostate cancer risk

The patrilineal patterns and their relationship with prostate cancer highlight the importance of ancestral burden in the development of complex diseases. Nevertheless, the sample size was the primary limitation because the present study attempts to answer an exploratory hypothesis that was secondary to the collected information. This impediment has decreased the accurate

estimation of the contribution of R1a lineage and the contribution to E1b1a/E1b1b and GHIJ lineages to PrCa; the statistical power for OR less or equal to that observed in these lineages was around 60% and 70%. Thus, the results of the present study should be confirmed using a larger sample size.

Even though, our results reinforced the contribution of family history and ethnic background to prostate cancer risk, many other factors have been associated with this pathology. Alterations in body mass index, glucose levels, cholesterol, triglycerides, and arterial pressure have shown a potential association with prostate cancer in Mexico [96, 97]. The contribution of smoking to PrCa risk is still not entirely clear, while high alcohol intake has also been related to increased PrCa risk [97]. All these factors are strongly present in the Mexican population and could potentially influence our results [98–101]. Howbeit, more studies should be conducted on Latin populations to reinforce our findings. The present study is the first of its kind conducted among Latins and stand out for the depth of its analyses.

The possible *Converso* founder effect could be clarified using high-fidelity markers in the haplogroup assignment, which might highlight the existence of Jewish lineages within the Mexican population and in other Latins. In addition, a relationship between the *BRCA1* and *BRCA2* and the high-fidelity lineages could reinforce our findings. Nonetheless, it is worth noting that at-risk lineages have also been reported in the Arabic and Lebanese populations, which have also contributed to the Mexican genetic architecture.

The former knowledge about the genetic architecture with autosomal and Y-chromosomal lineages from the Central Valley of Mexico population was one of the key strengths of the current study [8, 9, 14, 53]. The previous studies about the contribution of other genes in the sample explored herein broadens the landscape regarding the genetic influence on prostate cancer, and also strengthens our study [5, 26, 51, 97].

## Conclusion

Our results reinforced the contribution of family history and ethnic background to prostate cancer risk. Nonetheless, given its complex aetiology, the environmental, lifestyle, and xenobiotic exposure could be influencing our results. Also, our findings should be confirmed using a large sample size. Thus, our results should be interpreted in light of these limitations. The present study highlighted the use of genetic anthropology in studying complex medical conditions.

## Supporting information

**S1 Fig. Meta-analysis of the contribution of prostate cancer risk in the R1 carriers grouping the results by ethnicity using the random effects model.** OR, odds ratio; CI, confident intervals; $I^2$, proportion of the variance (heterogeneity). All data were obtained from several published studies.
(TIF)

**S2 Fig. Meta-analysis of the contribution of prostate cancer risk in the E carriers grouping the results by ethnicity using the random effects model.** OR, odds ratio; CI, confident intervals; $I^2$, proportion of the variance (heterogeneity). All data were obtained from several published studies.
(TIF)

**S3 Fig. Meta-analysis of the contribution of prostate cancer risk in the GIJ carriers grouping the results by ethnicity using the random effects model.** OR, odds ratio; CI, confident intervals; $I^2$, proportion of the variance (heterogeneity). All data were obtained from several

published studies.
(TIF)

**S4 Fig. Median-joining network of E1b1a and E1b1b lineages with 15 Y-STRs using the clinical manifestations as criterion.** Numbers in red represent the number of differences between one haplotype and other(s).
(TIF)

**S5 Fig. Median-joining network of G, I and J lineages with 15 Y-STRs using the clinical manifestations as criterion.** Numbers in red represent the number of differences between one haplotype and other(s).
(TIF)

**S6 Fig. Median-joining network of R1a lineage with 15 Y-STRs using the clinical manifestations as criterion.** Numbers in red represent the number of differences between one haplotype and other(s).
(TIF)

**S7 Fig. A multidimensional scale plot of $R_{ST}$ values of the E1b1b lineage with 6 Y-STRs using ethnicity as a criterion.** ASH, Ashkenazi Jews; BAR, Barcelona; CAT, Cataluña; CVMs (Central Valley of Mexico, data from Santana et al); CYP, Cyprus; FRA, France; GIR, Girona; ITA, Italy; LBN, Lebanon; LBY, Libya; LEVI, Levites Jews; LLE, Lleida; NL, Nuevo Leon; PSE, Palestinian Territory; SEPH, Sephardic Jews; TAR, Tarragona. All *p* values were adjusted with the method of false discovery rates. The dotted lines indicate that no significant differences were found among the populations.
(TIF)

**S8 Fig. A multidimensional scale plot of $R_{ST}$ values of the G lineage with 6 Y-STRs using ethnicity as a criterion.** AND, Andalusia; ANG, Anatolian Greeks; ARA, Aragon; ASH, Ashkenazi Jews; CAT, Cataluña; CHU; Chuetas; COR, Corsica; CVA; Valencian Community; CVMg (Central Valley of Mexico, data from Gomez et al); CVMs (Central Valley of Mexico, data from Santana et al); CYL, Castilla and Leon; GAL, Galicia; MEA, Middle East; NAV, Navarra; NL, Nuevo Leon; PVA, Basque Country; SEPH-IBP, Sephardic Jews from the Iberian Peninsula; SEPH-PRT, Sephardic Jews from Portugal; T&C, Turkish and Cypriots. All *p* values were adjusted with the method of false discovery rates. The dotted lines indicate that no significant differences were found among the populations.
(TIF)

**S9 Fig. A multidimensional scale plot of $R_{ST}$ values of the I lineage with 6 Y-STRs using ethnicity as a criterion.** AND, Andalusia; ANG, Anatolian Greeks; ARA, Aragon; ASH, Ashkenazi Jews; AST, Asturian Community; CAT, Cataluña; CHU; Chuetas; CLM, Castilla la Mancha; CVA; Valencian Community; CVMg (Central Valley of Mexico, data from Gomez et al); CVMs (Central Valley of Mexico, data from Santana et al); CYL, Castilla and Leon; EXT, Extremadura; FRA, France; GAL, Galicia; GTO, Guanajuato; ITA, Italy; MAD, Madrid; MEX, Mexico; MUR, Murcia; NAV, Navarra; NL, Nuevo Leon; PVA, Basque Country; RIO, La Rioja; SEPH-IBP, Sephardic Jews from the Iberian Peninsula; T&C, Turkish and Cypriots. All *p* values were adjusted with the method of false discovery rates. The dotted lines indicate that no significant differences were found among the populations.
(TIF)

**S10 Fig. A multidimensional scale plot of $R_{ST}$ values of the J lineage with 6 Y-STRs using ethnicity as a criterion.** ARA, Aragon; ASH, Ashkenazi Jews; CANT, Cantabria; CAT,

Cataluña; CHU; Chuetas; CLM, Castilla la Mancha; COHEN, Cohen Jews; COR; Corsica; CVA; Valencian Community; CVMg (Central Valley of Mexico, data from Gomez et al); CYL, Castilla and Leon; EXT, Extremadura; FRA, France; GTO, Guanajuato; ITA, Italy; LEVI, Levites Jews; MAD, Madrid; MAG, Maghreb; MAJ, Majorca; MUR, Murcia; NAFR, North Africa; NEA, Near East; NL, Nuevo Leon; PVA, Basque Country; RIO, La Rioja; SEPH-IBP, Sephardic Jews from the Iberian Peninsula; YEM, Yemen. All *p* values were adjusted with the method of false discovery rates. The dotted lines indicate that no significant differences were found among the populations.
(TIF)

**S11 Fig. A multidimensional scale plot of $R_{ST}$ values of the R1a lineage with 6 Y-STRs using ethnicity as a criterion.** AND, Andalusia; ARA, Aragon; ASH, Ashkenazi Jews; CAT, Cataluña; CHU; Chuetas; COHEN, Cohen Jews; CVA; Valencian Community; CVMg (Central Valley of Mexico, data from Gomez et al); CVMs (Central Valley of Mexico, data from Santana et al); CYL, Castilla and Leon; EXT, Extremadura; GAL, Galicia; ISR, Jews from Israel; MAD, Madrid; MAJ, Majorca; NL, Nuevo Leon; SEPH-IBP, Sephardic Jews from the Iberian Peninsula; SEPH-PRT, Sephardic Jews from Portugal. All *p* values were adjusted with the method of false discovery rates. The dotted lines indicate that no significant differences were found among the populations.
(TIF)

**S12 Fig. Median-joining network of Q-MEH2 lineage with 15 Y-STRs.** Numbers in red represent the number of differences between one haplotype and other(s).
(TIF)

**S13 Fig. Median-joining network of Q-L54 lineage with 15 Y-STRs.**
(TIF)

**S14 Fig. Median-joining network of Q-M3 lineage with 15 Y-STRs.**
(TIF)

**S1 Table. Haplotypes presented in cases and controls and its haplogroup (lineage) assignation with 17 Y-STRs using two different Bayesian predictors.** ID, identification of samples. Numbers from 1000 onwards represented prostate cancer cases. Only those haplogroups presenting assignation probabilities $\geq$ 0.70 were included in the analyses.
(XLSX)

**S2 Table. Y-STRs haplotypes from prior studies used to compare those haplotypes belonging to lineage E1b1a.** ID, identification of samples; NR, data not reported.
(XLSX)

**S3 Table. Y-STRs haplotypes from prior studies used to compare those haplotypes belonging to lineage E1b1b.** ID, identification of samples; NR, data not reported.
(XLSX)

**S4 Table. Y-STRs haplotypes from prior studies used to compare those haplotypes belonging to lineage G.** ID, identification of samples; NR, data not reported.
(XLSX)

**S5 Table. Y-STRs haplotypes from prior studies used to compare those haplotypes belonging to lineage I.** ID, identification of samples; NR, data not reported.
(XLSX)

**S6 Table. Y-STRs haplotypes from prior studies used to compare those haplotypes belonging to lineage J.** ID, identification of samples; NR, data not reported.
(XLSX)

**S7 Table. Y-STRs haplotypes from prior studies used to compare those haplotypes belonging to lineage R1a.** ID, identification of samples; NR, data not reported.
(XLSX)

**S8 Table. Y-STRs haplotypes from prior studies used to compare those haplotypes belonging to several lineages within the Mexican Mestizo population.** ID, identification of samples; NR, data not reported.
(XLSX)

**S9 Table. *RST* values estimated from 14 Y-STRs in those haplotypes belonging to haplogroup E1b1a.** All *p*-values were adjusted with the method of false discovery rates, those data presented in bold depicted significant values. ARAG, Arabian Gulf; CVMs, Central Valley of Mexico from Santana et al.; CYP, Cyprus; ITA, Italy; LBN; Lebanon; NL, Nuevo Leon; PRT, Portugal; UGA, Uganda; YUC, Yucatan.
(XLSX)

**S10 Table. *RST* values estimated from 14 Y-STRs in those haplotypes belonging to haplogroup E1b1b.** All *p*-values were adjusted with the method of false discovery rates, those data presented in bold depicted significant values. BAR, Barcelona; CAT, Cataluña; CVMs, Central Valley of Mexico from Santana et al.; CYP, Cyprus; FRA, France; GIR, Girona; ITA, Italy; LBN; Lebanon; LBY, Libya; LLE, Lleida; NL, Nuevo Leon; TAR, Tarragona.
(XLSX)

**S11 Table. *RST* values estimated from 6 Y-STRs in those haplotypes belonging to haplogroup E1b1b.** All *p*-values were adjusted with the method of false discovery rates, those data presented in bold depicted significant values. ASH-WEST, Ashkenazi Jews from west Europe; BAR, Barcelona; CAT, Cataluña; CVMs, Central Valley of Mexico from Santana et al.; CYP, Cyprus; FRA, France; GIR, Girona; ITA, Italy; JEWS-LEV, Levites Jews; JEWS-SEPH, Sephardim Jews; LBN; Lebanon; LBY, Libya; LLE, Lleida; NL, Nuevo Leon; PSE, Palestinian Territory TAR, Tarragona.
(XLSX)

**S12 Table. *RST* values estimated from 14 Y-STRs in those haplotypes belonging to haplogroup G.** All *p*-values were adjusted with the method of false discovery rates, those data presented in bold depicted significant values. AND, Andalusia; ARA, Aragon; ANG, Anatolian Greeks; CAT, Cataluña; CHU, Chuetas; COR, Corsica; CVA, *Comunitat* Valenciana; CVMg, Central Valley of Mexico from Gomez et al.; CVMs, Central Valley of Mexico from Santana et al.; CYL, Castilla y Leon; GAL, Galicia; NAV, Navarra; NL, Nuevo Leon; PVA, Basque Country; SEPH-PRT; Sephardim Jews from Portugal; T&C, Turkish-Cypriots.
(XLSX)

**S13 Table. *RST* values estimated from 6 Y-STRs in those haplotypes belonging to haplogroup G.** All *p*-values were adjusted with the method of false discovery rates, those data presented in bold depicted significant values. AND, Andalusia; ARA, Aragon; ANG, Anatolian Greeks; CAT, Cataluña; CHU, Chuetas; COR, Corsica; CVA, *Comunitat* Valenciana; CVMg, Central Valley of Mexico from Gomez et al.; CVMs, Central Valley of Mexico from Santana et al.; CYL, Castilla y Leon; GAL, Galicia; JEWS-ASH, Ashkenazi Jews; MEA, Middle East; NAV, Navarra; NL, Nuevo Leon; PVA, Basque Country; SEPH-IBP, Sephardim Jews from the

Iberian Peninsula; SEPH-PRT; Sephardim Jews from Portugal; T&C, Turkish-Cypriots.
(XLSX)

**S14 Table.** *RST* **values estimated from 14 Y-STRs in those haplotypes belonging to haplogroup I.** All *p*-values were adjusted with the method of false discovery rates, those data presented in bold depicted significant values. AND, Andalusia; ARA, Aragon; ANG, Anatolian Greeks; AST, Asturias; CAT, Cataluña; CHU, Chuetas; CLM, Castilla la Mancha; CVA, *Comunitat* Valenciana; CVMg, Central Valley of Mexico from Gomez *et al.*; CVMs, Central Valley of Mexico from Santana et al.; CYL, Castilla y Leon; EXT, Extremadura; FRA, France; GAL, Galicia; GTO, Guanajuato; ITA, Italy; MAD, Madrid; MUR, Murcia; NAV, Navarra; NL, Nuevo Leon; PVA, Basque Country; RIO, La Rioja; SEPH-PRT; Sephardim Jews from Portugal; T&C, Turkish-Cypriots.
(XLSX)

**S15 Table.** *RST* **values estimated from 6 Y-STRs in those haplotypes belonging to haplogroup I.** All *p*-values were adjusted with the method of false discovery rates, those data presented in bold depicted significant values. AND, Andalusia; ARA, Aragon; ANG, Anatolian Greeks; AST, Asturias; CAT, Cataluña; CHU, Chuetas; CLM, Castilla la Mancha; CVA, *Comunitat* Valenciana; CVMg, Central Valley of Mexico from Gomez *et al.*; CVMs, Central Valley of Mexico from Santana et al.; CYL, Castilla y Leon; EXT, Extremadura; FRA, France; GAL, Galicia; GTO, Guanajuato; ITA, Italy; JEWS-ASH, Ashkenazi Jews; MAD, Madrid; MUR, Murcia; NAV, Navarra; NL, Nuevo Leon; PVA, Basque Country; RIO, La Rioja; SEPH-IBP; Sephardim Jews from the Iberian Peninsula; T&C, Turkish-Cypriots.
(XLSX)

**S16 Table.** *RST* **values estimated from 14 Y-STRs in those haplotypes belonging to haplogroup J.** All *p*-values were adjusted with the method of false discovery rates, those data presented in bold depicted significant values. ARA, Aragon; CAT, Cataluña; CHU, Chuetas; CLM, Castilla la Mancha; COR, Corsica; CVA, *Comunitat* Valenciana; CVMg, Central Valley of Mexico from Gomez et al.; CYL, Castilla y Leon; CANT, Cantabria; EXT, Extremadura; FRA, France; GTO, Guanajuato; ITA, Italy; MAD, Madrid; MAJ, Majorca; MUR, Murcia; NL, Nuevo Leon; PVA, Basque Country; RIO, La Rioja.
(XLSX)

**S17 Table.** *RST* **values estimated from 6 Y-STRs in those haplotypes belonging to haplogroup J.** All *p*-values were adjusted with the method of false discovery rates, those data presented in bold depicted significant values. ARA, Aragon; CAT, Cataluña; CHU, Chuetas; CLM, Castilla la Mancha; COR, Corsica; CVA, *Comunitat* Valenciana; CVMg, Central Valley of Mexico from Gomez et al.; CYL, Castilla y Leon; CANT, Cantabria; EXT, Extremadura; FRA, France; GTO, Guanajuato; ITA, Italy; JEWS-ASH, Ashkenazi Jews; JEWS-COHEN, Cohen Jews; JEWS-LEVI, Levite Jews; MAD, Madrid; MAJ, Majorca; MUR, Murcia; MAGH, Maghreb; NEA, Near East; NL, Nuevo Leon; NAFR, North Africa; MAJ, Majorca; PVA, Basque Country; RIO, La Rioja; SEPH-IBP; Sephardim Jews from the Iberian Peninsula; YEM, Yemen.
(XLSX)

**S18 Table.** *RST* **values estimated from 14 Y-STRs in those haplotypes belonging to haplogroup R1a.** All *p*-values were adjusted with the method of false discovery rates, those data presented in bold depicted significant values. AND, Andalusia; ARA, Aragon; CAT, Cataluña; CHU, Chuetas; CVA, *Comunitat* Valenciana; CVMg, Central Valley of Mexico from Gomez et al.; CVMs, Central Valley of Mexico from Santana et al.; CYL, Castilla y Leon; EXT,

Extremadura; GAL, Galicia; JEWS-ASH, Ashkenazi Jews; JEWS-COHEN, Cohen Jews; JEWS-IBP, Jews from the Iberian Peninsula; JEWS-ISR, Jews from Israel; MAD, Madrid; MAJ, Majorca; NL, Nuevo Leon; SEPH-PRT; Sephardim Jews from Portugal; T&C, Turkish and Cypriots.
(XLSX)

**S19 Table.** *RST* **values estimated from 6 Y-STRs in those haplotypes belonging to haplogroup R1a.** All *p*-values were adjusted with the method of false discovery rates, those data presented in bold depicted significant values. AND, Andalusia; ARA, Aragon; CAT, Cataluña; CHU, Chuetas; CVA, *Comunitat* Valenciana; CVMg, Central Valley of Mexico from Gomez et al.; CVMs, Central Valley of Mexico from Santana et al.; CYL, Castilla y Leon; EXT, Extremadura; GAL, Galicia; JEWS-ASH, Ashkenazi Jews; JEWS-COHEN, Cohen Jews; JEWS-IBP, Jews from the Iberian Peninsula; JEWS-ISR, Jews from Israel; MAD, Madrid; MAJ, Majorca; NL, Nuevo Leon; SEPH-IBP; Sephardim Jews from the Iberian Peninsula; SEPH-PTG; Sephardim Jews from Portugal; T&C, Turkish and Cypriots.
(XLSX)

## Acknowledgments

We would like to thank the participation of the men from the several institutions; their collaboration made this study possible. CONACYT supported the master's in science studies of EA-T through a scholarship. We also thank Vanessa Morillón-Torres for her uninterested help in the determination of some haplogroups.

## Author Contributions

**Conceptualization:** Luisa E. Torres-Sánchez, Rocío Gómez.

**Data curation:** Luisa E. Torres-Sánchez, Esther A. Hernández-Tobías, David Véliz, Rocío Gómez.

**Formal analysis:** Luisa E. Torres-Sánchez, Esther A. Hernández-Tobías, David Véliz, Rocío Gómez.

**Funding acquisition:** Luisa E. Torres-Sánchez, Rocío Gómez.

**Investigation:** Esmeralda Álvarez-Topete, Luisa E. Torres-Sánchez, David Véliz, Marco Antonio Meraz-Ríos, Rocío Gómez.

**Methodology:** Esmeralda Álvarez-Topete, Luisa E. Torres-Sánchez, David Véliz, Jesús G. Hernández-Pérez, Ma. de Lourdes López-González, Marco Antonio Meraz-Ríos, Rocío Gómez.

**Project administration:** Rocío Gómez.

**Resources:** Luisa E. Torres-Sánchez, Rocío Gómez.

**Software:** Esther A. Hernández-Tobías, David Véliz, Rocío Gómez.

**Supervision:** Luisa E. Torres-Sánchez, Ma. de Lourdes López-González, Rocío Gómez.

**Validation:** Rocío Gómez.

**Visualization:** Rocío Gómez.

**Writing – original draft:** Rocío Gómez.

**Writing – review & editing:** Luisa E. Torres-Sánchez, Esther A. Hernández-Tobías, David Véliz, Jesús G. Hernández-Pérez, Marco Antonio Meraz-Ríos, Rocío Gómez.

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
