## [Decision Letter · Decision Letter 0]

17 Mar 2024

PONE-D-23-43139Circum-Mediterranean Influence in the Y-Chromosome Lineages Associated with Prostate Cancer in Mexican Men: A Converso Heritage Founder Effect?PLOS ONE

Dear Dr. Gomez,

Thank you for submitting your manuscript to PLOS ONE. After careful consideration, we feel that it has merit but does not fully meet PLOS ONE’s publication criteria as it currently stands. Therefore, we invite you to submit a revised version of the manuscript that addresses the points raised during the review process.

The manuscript is interesting. However, many of the arguments are not justified. Please consider the questions from both of the reviewers and revised the manuscript accordingly. Please ensure that your decision is justified on PLOS ONE’s publication criteria and not, for example, on novelty or perceived impact.

We look forward to receiving your revised manuscript.

Kind regards,

Gyaneshwer Chaubey

Academic Editor

PLOS ONE

Journal Requirements:

4. Thank you for stating the following in the Acknowledgments Section of your manuscript: "We would like to thank the participation of the men from the several institutions; their collaboration made this study possible. Funding for this study was provided by CONACYT grants 261268 and 178239 (RG), and 140482 and 272810 (LETS). CONACYT also supported the master’s in science studies of EA-T through a scholarship. We also thank to Vanessa Morillón-Torres and Opata Edward Kwame, M. Sc., for their disinterested help in the determination of some haplogroupsband proofreading, respectively. "

Please remove any funding-related text from the manuscript and let us know how you would like to update your Funding Statement. Currently, your Funding Statement reads as follows: "The author(s) received no specific funding for this work. "

Reviewers' comments:

Reviewer's Responses to Questions

**Comments to the Author**

1. Is the manuscript technically sound, and do the data support the conclusions?

Reviewer #1: Yes

Reviewer #2: Yes

2. Has the statistical analysis been performed appropriately and rigorously? 

Reviewer #1: I Don't Know

Reviewer #2: I Don't Know

3. Have the authors made all data underlying the findings in their manuscript fully available?

Reviewer #1: Yes

Reviewer #2: Yes

4. Is the manuscript presented in an intelligible fashion and written in standard English?

Reviewer #1: Yes

Reviewer #2: Yes

5. Review Comments to the Author

Reviewer #1: It is an interesting study. The paper need some improvement in the language.

Although R1a haplogroup as per statistical tests used is claimed to contribute to prostate cancer (OR adjusted: 8.04, 95%CI 1.41 – 45.80) on perusing raw data an extremely small number of cases and controls are found in the study cohort. This makes it biologically less meaningful and the limitation should be discussed in the discussion and also indicated in the abstract - otherwise a wrong idea will be conveyed to the readers especially in view of high OR.

Line 51 - Sentence not clear due to the word incidence

Line 91 - replace the word statemented with a better word

Line 168 - replace the word unknew with a better word

Line 182 - correct the word avid

Line 197 to 200 - write better - many uses of the word "those"

Line 595 - correct the word technician

Line 605 - replace the word disinterested with a more appropriate word

Reviewer #2: In the manuscript “Circum-Mediterranean Influence in the Y-Chromosome Lineages Associated with Prostate Cancer in Mexican Men: A Converso Heritage Founder Effect?” the authors analyze the paternal origin of 152 patients with prostate cancer (cases) and 372 population (controls) from the Mexican Mestizo population. They focus on the influence of ethnicity in prostate cancer. The subject is the great interest for male health. The genetic factors highly contributes to the genesis of prostate cancer and ethnic may influence this contribution

However to my point of view some issues may be improved. My main concern is regarding the low number of cases. It looks as the authors do not statistically analyzed the needed number to perform the study. They do not add this important information in the statistic section.

Introduction section:

Line 237 define CVM please

Results

In table 5, why it should be different risk between ethnicity and differences grades of prostate cancer? Is it a biological or cultural effect? I am concerned about the low number of prostate cancer Gleason <7 N=37

Discussion section

It is an interesting description of ethnicity. I should suggest to focus on prostate cancer more than to the ethnicity, namely to focus in the aim of the work.

Regarding other risk factors of prostate cancer, I agree with the authors (as they mentioned in the conclusion section) that is a limitation of the study, is it possible to obtain more data from the patients? Anyway it is an important limitation that should be discussed more deeply and it does not represent a conclusion of the study.

Please explain “ The differences found between cases (23.5%) and controls

500 (13.1%) in the Central-East region could be explained by the exposed causes in prior paragraphs “ (line 500) How the exposed causes explain the differences in prostate cancer? .

References section

I could not find the reference Martinez- Cortes.

6. PLOS authors have the option to publish the peer review history of their article (what does this mean?). If published, this will include your full peer review and any attached files.

Reviewer #1: **Yes: **Kamani Tennekoon

Reviewer #2: **Yes: **Rossana Sapiro

---

## [Author Response · Author response to Decision Letter 0]

23 May 2024

Response to Editor and Reviewer comments

We would like to thank the Editor and Reviewers for the detailed review of our manuscript and the constructive feedback provided on it. We have carefully evaluated the points raised in the review and made the corresponding changes to the text. These changes have significantly improved the quality of the manuscript and appear highlighted in yellow in the main text. Our responses to the Editor and the Reviewer one appear in blue font below; Reviewer two appears in red font.

Response to Editor

We note that Figure 1 in your submission contain [map/satellite] images which may be copyrighted. All PLOS content is published under the Creative Commons Attribution License (CC BY 4.0), which means that the manuscript, images, and Supporting Information files will be freely available online, and any third party is permitted to access, download, copy, distribute, and use these materials in any way, even commercially, with proper attribution. For these reasons, we cannot publish previously copyrighted maps or satellite images created using proprietary data, such as Google software (Google Maps, Street View, and Earth). We require you to either (1) present written permission from the copyright holder to publish these figures specifically under the CC BY 4.0 license, or (2) remove the figures from your submission.

We appreciate this suggestion. Nonetheless, the map used as Figure 1 of our manuscript is not a copyrighted map. Given that we need some specifications to depict particular features, I create totally this figure. In addition, we confirm that the map used in such figure has not been used or copyrighted, previously. On behalf of my co-authors, as well as on my behalf, we have included a letter confirming the sentences mentioned before.

Response to Reviewers’

Reviewer #1

1. It is an interesting study. The paper needs some improvement in the language.

Thank you for this observation. This new version has been proofread by an English Native.

2. Although R1a haplogroup as per statistical tests used is claimed to contribute to prostate cancer (OR adjusted: 8.04, 95%CI 1.41 – 45.80) on perusing raw data an extremely small number of cases and controls are found in the study cohort. This makes it biologically less meaningful, and the limitation should be discussed in the discussion and also indicated in the abstract - otherwise a wrong idea will be conveyed to the readers especially in view of high OR.

We appreciate these suggestions, which have been attended at whole. The Discussion section includes the arguments about the effect of the sample size, a calculous of the statistical power, and the suggestion to confirm our findings. Similar arguments have been included in the Abstract section. 

3. Line 51 - Sentence not clear due to the word incidence.

We apologise for this mistake. Following your accurate observation, we have included the word “incidence” in the first sentence of the Introduction section. 

4. Line 91 - replace the word statemented with a better word.

We appreciate this suggestion. Thus, we have replaced “statement” with “suggested” to indicate that such population interaction was likely, diminishing the force of the pronouncement. 

5. Line 168 - replace the word unknew with a better word.

We thank the reviewer for this suggestion. We replaced “unknew” by “did not know”. 

6. Line 182 - correct the word avid.

We apologize for this typographic error; it has been corrected. 

7. Line 197 to 200 - write better - many uses of the word "those".

In full agreement with your comment. We have modified the redaction of this paragraph for accuracy and clarity in its reading.

8. Line 595 - correct the word technician.

We agree with your comment; the word technician analyses was replaced by technical procedures.

9. Line 605 - replace the word disinterested with a more appropriate word.

Thank you for this observation. We replaced disinterested by uninterested.

Reviewer #2

In the manuscript “Circum-Mediterranean Influence in the Y-Chromosome Lineages Associated with Prostate Cancer in Mexican Men: A Converso Heritage Founder Effect?” the authors analyze the paternal origin of 152 patients with prostate cancer (cases) and 372 population (controls) from the Mexican Mestizo population. They focus on the influence of ethnicity in prostate cancer. The subject is the great interest for male health. The genetic factors highly contribute to the genesis of prostate cancer and ethnic may influence this contribution. However, to my point of view some issues may be improved. 

1. My main concern is regarding the low number of cases. It looks as the authors do not statistically analyzed the needed number to perform the study. They do not add this important information in the statistic section.

We appreciate your suggestions. We have replicated all your comments in the Discussion section, explaining the several arguments about your justified concerns.

2. Introduction section.

Line 237 define CVM please.

We appreciate this commentary. We have included the mean of the CVM acronym.

3. Results section

In table 5, why it should be different risk between ethnicity and differences grades of prostate cancer? Is it a biological or cultural effect? I am concerned about the low number of prostate cancer Gleason <7 N=37

We thank the reviewer for this point. We have attended and replicated your comments in a Discussion section in a broad context.

4. Discussion section

It is an interesting description of ethnicity. I should suggest to focus on prostate cancer more than to the ethnicity, namely to focus in the aim of the work. 

We thank the reviewer for this suggestion. As you mentioned we have focused on the influence of ethnicity in prostate cancer. In former reports our research group have discussed about the genetic contribution of several genes (PMID:32219892, 30774776, 27193223), the environmental pressure (PMID:34520388), and the concomitant diseases to prostate cancer development. In the present study, our goal was to determine whether paternal ancestral lineages have certain contribution to prostate cancer given that the family history is critical in its developed. Our findings obligate us to include the ethnicity as a cornerstone of the discussion. First at all, because the lineages found are not natives of the original populations of Mexico suggesting a possibly founder effect, which is novel in our research. Rather, is impossible to separate the study of uniparental markers such as Y-chromosome out of the historical, cultural and religious contexts. We acknowledge for its observation, but we think that in the literature exist many papers regarding with the genetic influence of cancer. Using the keyword of “prostate cancer” AND “genetics” there are 42,438 results. By contrast, using the keywords “prostate cancer” AND “Y-chromosome” there are 107 results. Of these, only seven were related to the influence haplogroups or lineages in the prostate cancer. Nonetheless, attending your commentary we have added a section (A medical context) explaining the possible mechanisms that could be involved in the possible contribution of patrilineages in the prostate cancer. This section is previous the Historical context. In addition, we have added some subheadings to be more clear in the Discussion. 

5. Regarding other risk factors of prostate cancer, I agree with the authors (as they mentioned in the conclusion section) that is a limitation of the study, is it possible to obtain more data from the patients? Anyway it is an important limitation that should be discussed more deeply and it does not represent a conclusion of the study. Please, explain “The differences found between cases (23.5%) and controls

500 (13.1%) in the Central-East region could be explained by the exposed causes in prior paragraphs “(line 500) How the exposed causes explain the differences in prostate cancer? 

We acknowledge all your suggestions; we agree totally with your points. 

About the possibility to obtain more data from the patients, several papers have been published previously (PMID:32219892, 30774776, 27193223, 34520388). Regarding the limitations, these have been discussed broadly and we have modified the Conclusion. Also, we have explained the differences found in the Central-East region. 

6. References section

I could not find the reference Martinez- Cortes.

You are right. Thank you for pointing out this omission. We have included this reference.

---

## [Decision Letter · Decision Letter 1]

17 Jul 2024

Circum-Mediterranean Influence in the Y-Chromosome Lineages Associated with Prostate Cancer in Mexican Men: A Converso Heritage Founder Effect?

PONE-D-23-43139R1

Dear Dr. Gomez,

We’re pleased to inform you that your manuscript has been judged scientifically suitable for publication and will be formally accepted for publication once it meets all outstanding technical requirements.

Kind regards,

Gyaneshwer Chaubey

Academic Editor

PLOS ONE

Additional Editor Comments (optional):

Reviewers' comments:

Reviewer's Responses to Questions

**Comments to the Author**

1. If the authors have adequately addressed your comments raised in a previous round of review and you feel that this manuscript is now acceptable for publication, you may indicate that here to bypass the “Comments to the Author” section, enter your conflict of interest statement in the “Confidential to Editor” section, and submit your "Accept" recommendation.

Reviewer #1: All comments have been addressed

Reviewer #2: All comments have been addressed

2. Is the manuscript technically sound, and do the data support the conclusions?

Reviewer #1: Yes

Reviewer #2: Yes

3. Has the statistical analysis been performed appropriately and rigorously? 

Reviewer #1: Yes

Reviewer #2: Yes

4. Have the authors made all data underlying the findings in their manuscript fully available?

Reviewer #1: Yes

Reviewer #2: Yes

5. Is the manuscript presented in an intelligible fashion and written in standard English?

Reviewer #1: Yes

Reviewer #2: Yes

6. Review Comments to the Author

Reviewer #1: (No Response)

Reviewer #2: The authors have correctly addressed all my comments. The manuscript has been improved after the corrections.

7. PLOS authors have the option to publish the peer review history of their article (what does this mean?). If published, this will include your full peer review and any attached files.

Reviewer #1: No

Reviewer #2: No

---

## [Editor Report · Acceptance letter]

8 Aug 2024

PONE-D-23-43139R1 

PLOS ONE

Dear Dr. Gomez, 

I'm pleased to inform you that your manuscript has been deemed suitable for publication in PLOS ONE. Congratulations! Your manuscript is now being handed over to our production team.

Kind regards, 

on behalf of

Gyaneshwer Chaubey 

Academic Editor

PLOS ONE